# A unified European hydrogen infrastructure planning to support the rapid scale-up of hydrogen production

Ioannis Kountouris [1] ✉, Rasmus Bramstoft[1], Theis Madsen[1], Juan Gea-Bermúdez [2], Marie Münster [1] & Dogan Keles[1]

Hydrogen will become a key player in transitioning toward a net-zero energy system. However, a clear pathway toward a unified European hydrogen infrastructure to support the rapid scale-up of hydrogen production is still under discussion. This study explores plausible pathways using a fully sector-coupled energy system model. Here, we assess the emergence of hydrogen infrastructure build-outs connecting neighboring European nations through hydrogen import and domestic production centers with Western and Central European demands via four distinct hydrogen corridors. We identify a potential lock-in effect of blue hydrogen in the medium term, highlighting the risk of long-term dependence on methane. In contrast, we show that a self-sufficient Europe relying on domestic green hydrogen by 2050 would increase yearly expenses by around 3% and require 518 gigawatts of electrolysis capacity. This study emphasizes the importance of rapidly scaling up electrolysis capacity, building hydrogen networks and storage facilities, deploying renewable electricity generation, and ensuring coherent coordination across European nations.

The European Commission (EC) has continuously revised its renewable hydrogen strategy since launching the "Clean Energy for All Europeans" initiative in November 2016[1], with a series of increasingly ambitious targets. Recently, in July 2021, the EC introduced the "Fit-for-55" proposal, which aims to reduce the European Union's greenhouse gas emissions by 55% and established a target of achieving 6.7 million tons of renewable domestic hydrogen production by 2030[2]. In May 2022, the domestic target was updated by the latest "REPowerEU" communication actions in response to the disruption of the global energy market, aiming to decrease dependence on Russian fossil fuels and accelerate the shift toward renewable energy sources in the European Union[3]. The target has been set at 10 million tonnes or 333 terawatt-hours (TWh) of domestic hydrogen production and an additional 10 million tonnes of imports. In addition, most of the European countries are announcing national strategies toward 2030 for installed electrolyzer capacity[4,5]. Although short to medium-term hydrogen strategies are announced on a national and European scale, a clear path toward a unified European hydrogen infrastructure in a future sector-coupled energy system is still under discussion.

First, the European gas industry, in a series of reports[5–8], expresses the European Hydrogen Backbone (EHB) vision consisting of a hydrogen network composed from new pipelines and repurposed the existing natural gas transmission network. Despite the fact that they demonstrate enormous benefits of connecting demand and production centers by exploiting increasingly redundant natural gas infrastructure, their methodology is limited to the vision and strategies of the national gas and electricity transmission system operators. Second, while recent studies focus on detailed self-sufficient carbon-neutral European energy system scenarios for 2050, they have not integrated hydrogen networks[9] and focus mainly on the benefits of sector coupling[10]. Third, few recent studies have attempted to use techno-economic optimization methodologies to analyze the value of hydrogen infrastructure in Europe. Some focus on hydrogen grid development in 2050[11], by assessing multiple weather data[12] or

[1]Department of Technology, Management and Economics, Technical University of Denmark, Produktionstorvet, Bygning 424, Kongens Lyngby, Denmark.
[2]Joint Research Centre (JRC), European Commission, Calle Inca Garcilaso, 3, Sevilla, Spain. ✉e-mail: iokoun@dtu.dk

examining both hydrogen and power grid trade-offs[13,14] yet neglecting pathway dependencies. Fourth, while providing an energy transition pathway analysis, some studies do not include blue hydrogen as a production alternative[15,16]. Fifth, comprehensive studies are carried out, yet they disregard the interaction of electricity and hydrogen sectors with the heating sector[17,18] or while modeling a sector-coupled European energy system, they focus on a specific country or region only. For example, they investigate the role of offshore wind in the North Sea[15], assess if offshore hydrogen production is more competitive than onshore[19] or examine scenarios for designing a net-zero energy system and hydrogen supply pathways to greenhouse gas neutrality in Germany[20,21]. Finally, some investigate alternative transition pathways for the European energy system, including electricity and hydrogen infrastructure development[22] yet assuming self-sufficiency. A few optimization studies[23,24] take into account importing hydrogen from other nations, but they do not examine the impact on the European hydrogen economy and regional infrastructure. A deep decarbonization of not only the power sector but also an advanced sector coupling through least-cost-optimal and no-regrets decisions is shown to result in the most significant emission reduction[25,26]. Despite the ambitious plans for scaling up the future hydrogen economy and infrastructure, a comprehensive energy transition pathway analysis based on a techno-economic optimization of a sector-coupled European energy system is lacking.

In this study, we explore plausible pathways for establishing an integrated European hydrogen infrastructure to support the rapid scale-up of hydrogen production while considering possible import opportunities as well as synergies between energy vectors and sectors across Europe. We thereby seek to provide answers to three fundamental questions. First, what are the main trends in where, when, and how to produce hydrogen at different European locations in the future, considering synergies and interactions across the entire energy system? Second, what might the role of blue hydrogen be in the energy transition, and how does it affect the European energy system? Third, what is the impact of a self-sufficient European hydrogen economy exclusively based on domestic green hydrogen production without hydrogen import opportunities?

To address each research question, we design three distinct, yet plausible scenarios. 1) The Hydrogen Europe (H2E) scenario allows the import of green hydrogen and competition between different hydrogen production technologies, i.e., electrolyzers, Steam Methane Reforming (SMR), and SMR-CCS (Carbon Capture and Storage CCS). 2) The Green Hydrogen Europe (GH2E) scenario also permits the import of green hydrogen but only allows electrolyzers to produce hydrogen due to the potential environmental risks and methane dependence of blue hydrogen. 3) The Self-Sufficient Green Hydrogen Europe (SSGH2E) scenario only allows European electrolyzers to produce hydrogen, i.e., no import and no SMR-CCS technologies. For additional scenario information, details, and motivation, see the section Scenario Choice and Description.

This study uses the terms 'green,' 'gray,' and 'blue' hydrogen to reflect the technology-specific production. Green hydrogen is generated by water electrolysis using renewable or low-carbon energy electricity (see more details in Supplementary Method 4). Gray is derived from steam methane reforming (SMR) using natural gas. Blue hydrogen is produced from natural gas using SMR with carbon capture and storage (CCS), reducing production emissions.

We implement the scenarios in the European energy system model, Balmorel[19,27], accounting for cross-regional network expansion and multiple supply options to meet regional demands by minimizing the investment and operational expenses of the total energy system. Additional sensitivities, such as alternative spatial allocation of hydrogen derivatives demand or the possibility of shipping imports, are also assessed. In contrast with other greenfield studies, our analysis assesses energy transition pathways sequentially every five years,

simulating the European energy system (EU 27, the UK, Norway, Switzerland, and the Balkan countries) to provide an optimal expansion pathway and long-term planning. Figure 1 illustrates the overview of the energy vectors and sectors covered in the model.

The main results of the analysis indicate a consistent advantage in the economic feasibility of hydrogen production facilities on the continent's periphery while simultaneously linking them to the hydrogen demand of Western and Central Europe through a combination of new and repurposed infrastructure. Blue hydrogen is a transition fuel for the hydrogen economy, although it has a high lock-in effect and increases the dependency on natural gas after 2035. By early 2040, a hydrogen economy without blue hydrogen requires faster grid and storage expansion, additional quantities of hydrogen import, and additional renewable investments. Technological uncertainty such as potential future low $CO_2$ capture rates and increasing $CO_2$ transportation and storage costs can accelerate the green hydrogen investments. Hydrogen imports will moderately impact the energy system in 2050 but not in 2030. A self-sufficient and green hydrogen economy in Europe increases system costs by only 3% and requires around 507 GW of water electrolysis, which is ~127 times greater than the European installed capacity 4 gigawatts (GW) in 2022[28].

## Results

### Hydrogen production from short-term to long-term perspective
To supply the estimated hydrogen demand, we find Europe's electrolyzer capacity ranging from 24 GW (73 TWh) to 73 GW (320 TWh) by 2030, and 310 GW (989 TWh) to 518 GW (1708 TWh) by 2050, depending on the scenario (see Fig. 2a, with values in parentheses denoting the corresponding production levels). In general, Fig. 3 visualizes a significant deployment of electrolyzer capacity in the South European countries toward 2050, due to the declining investment costs of solar photovoltaic (solar PV) and the high renewable potential, combined with the flexible operation of the electrolysis technology and potential underground hydrogen storage. Furthermore, countries with excellent onshore and offshore wind potentials (Supplementary Fig. 6), have the potential to become leaders in hydrogen production as well. Toward 2050, a well-connected hydrogen grid will emerge and create hydrogen corridors connecting production and demand centers. However, significant hydrogen demands for steel, process heat, transportation, ammonia, and high-value chemicals, are frequently concentrated in European regions/countries with lower renewable energy resources.

Our results shed light on the competition between gray, blue, and green hydrogen production pathways. Gray hydrogen is out-competed based on the $CO_2$ taxation projection (€ 150 $tCO_2^{-1}$, by 2030) implemented across the scenarios. Consequently, existing conventional SMR capacities may become stranded assets before their operational lifetimes are reached. Both electrolysis and SMR-CCS hydrogen generation capacities are being deployed in the H2E scenario with almost similar production cost marginals in the short to medium term, resulting in installed SMR-CCS capacities of 21 GW by 2030 and 62 GW by 2050 in the H2E scenario (see Fig. 2a).

### Blue hydrogen and its potential lock-in effect
Although electrolytic hydrogen becomes the leading technology deployed in the long-term, blue hydrogen acts as a transition fuel in the short to medium term, potentially creating lock-in effects with the risk of long-term dependence on natural gas, as shown in Fig. 2a.

While natural gas prices are falling toward 2035 due to decreased natural gas demand across all sectors, green hydrogen's relatively higher costs remain a result of endogenously formed electricity prices and techno-economic specifications. Furthermore, we observe a steady operation of the technical assets for blue hydrogen production, which makes it capable of providing base-load services. As a result, domestic blue hydrogen production

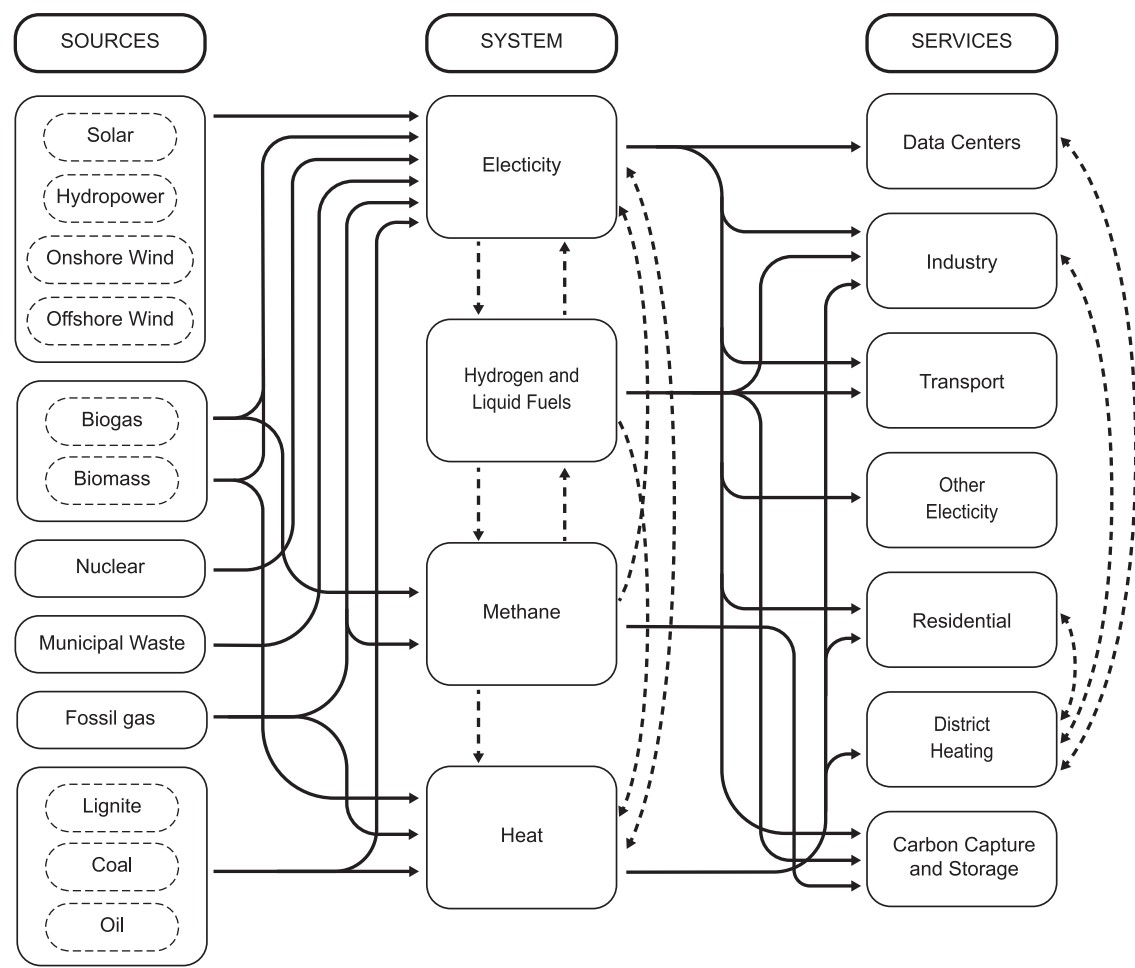

**Fig. 1 | Schematic overview of energy vectors and sector coupling modeling in Balmorel.** Balmorel is a partial equilibrium model that converts energy sources into multiple energy vectors, including electricity, methane, heat, and hydrogen. These vectors deliver energy services across various sectors, defined as an exogenous final demand. The model includes the option to invest in and utilize Carbon Capture and Storage (CCS), thereby lowering the overall final emission intensity when converting conventional sources, such as fossil gas, into energy vectors.

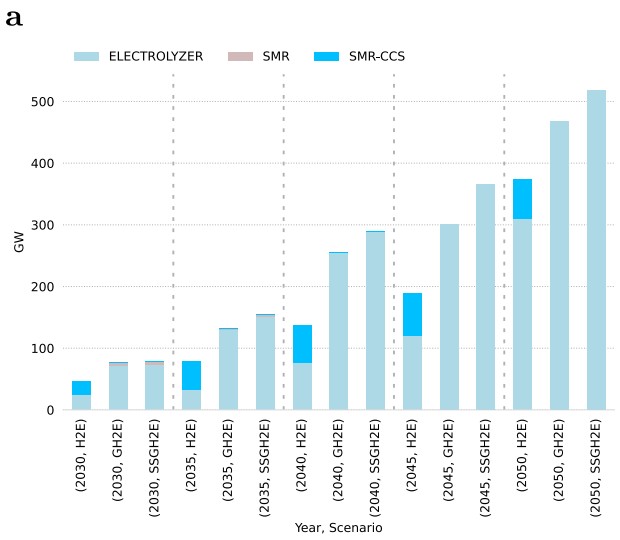

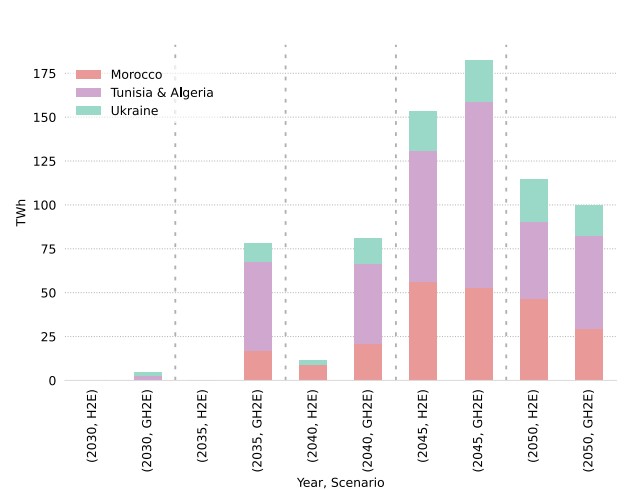

**Fig. 2 | Evolution of the aggregated hydrogen production installed capacities (GW) across the three scenarios and optimal imported hydrogen quantities from three countries and entry points: Tunisia & Algeria - Italy, Morocco-Spain, Ukraine - Slovakia. a** Gray hydrogen based on Steam Methane Reforming (SMR) phases out due to the high CO$_2$ taxation projection. A lock-in effect of blue hydrogen produced through SMR with Carbon Capture and Storage (SMR-CCS) is observed in the Hydrogen Europe (H2E) scenario. The Green Hydrogen Europe (GH2E) and Self-Sufficient Green Hydrogen Europe (SSGH2E) scenarios illustrate the outcomes when blue hydrogen deployment is not permitted. **b** The current hydrogen export national strategies could cover ~20% of the domestic hydrogen demand in 2030, as assumed in this study. By 2050, the overall target volumes in terawatt-hours (TWh) are 375 TWh, 115 TWh, and 100 TWh, respectively, which potentially satisfy up to 33% of the European domestic hydrogen demand. Scenario SSGH2E is not shown because trade with neighboring countries is not allowed.

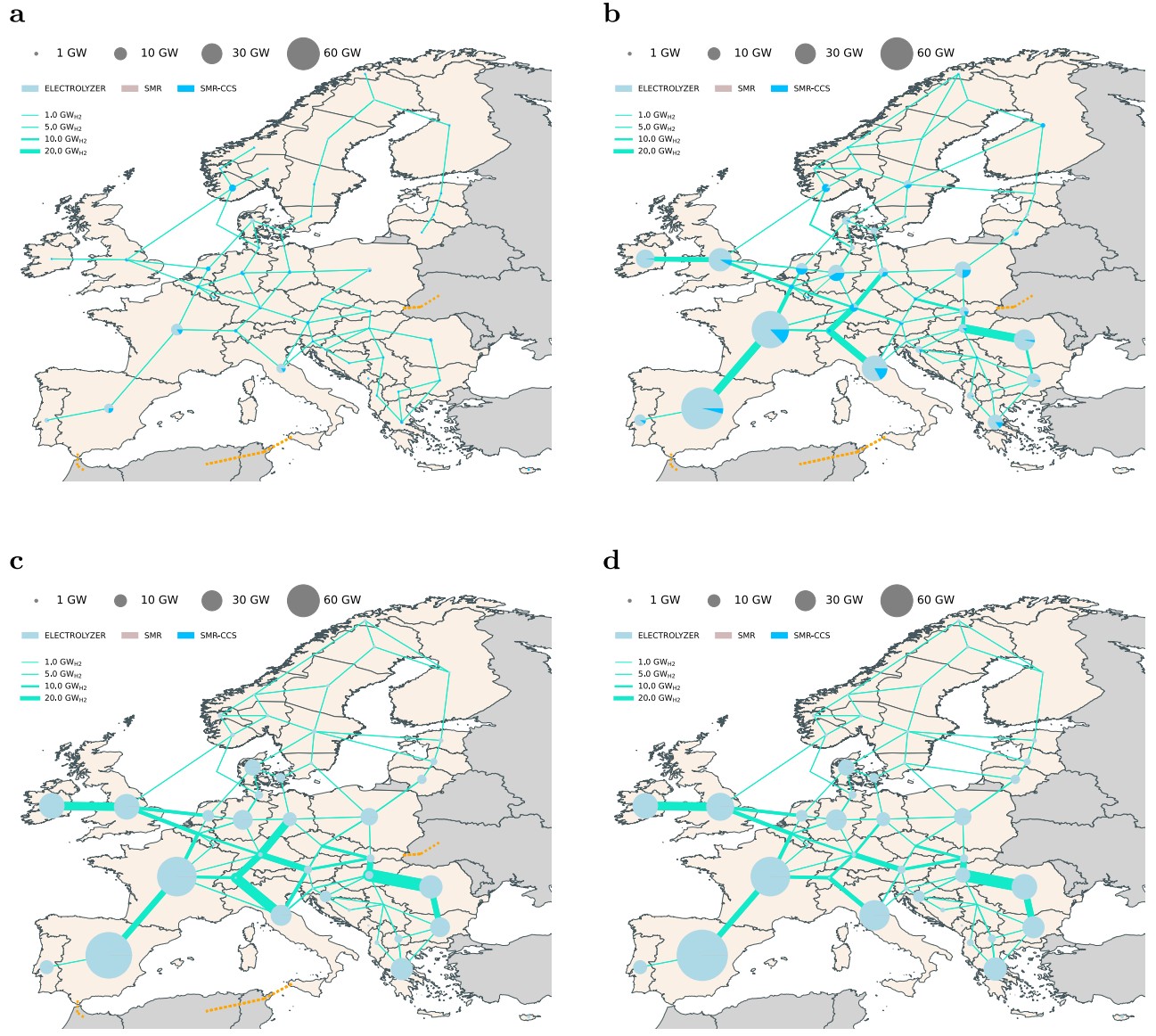

**Fig. 3 | Hydrogen-optimized grid and production centers.** The circle size represents the total installed regional or national hydrogen capacities in gigawatt (GW), consisting of three potential technologies: Electrolyzer, Steam Methane Reforming (SMR), and Steam Methane Reforming with Carbon Capture and Storage (SMR-CSS). The line widths indicate the optimal capacity of the pipeline. The orange dashed line indicates prospective hydrogen import connection scenarios.

**a** Optimal allocation of hydrogen production centers and grid development by 2030 according to the Hydrogen Europe (H2E) scenario. **b**–**d** Three plausible development scenarios for the European hydrogen production centers and infrastructure by 2050. **b** H2E **c** Green Hydrogen Europe (GH2E) **d** Self-Sufficient Green Hydrogen Europe (SSGH2E). Figures depicting the 2030 to 2045 evolution are available in the Supplementary Figs.

increases, reducing the requirement for hydrogen network expansion and storage.

While blue hydrogen constitutes a considerable amount due to its lock-in during the short to medium term, its natural gas usage is within the domestic resources with the European countries[29], which ensures self-sufficiency and thereby eliminates the imports from third-countries (see further Supplementary Discussion 1).

Due to the debate about classifying blue hydrogen produced by SMR-CCS as low carbon fuel, encountering the risks of leakages and potential lock-in effects, we show that a system without blue hydrogen would require additional water electrolysis investments of ~48 GW by 2030 and 157 GW by 2050, compared to the H2E findings (see Fig. 2a).

### Competitiveness of European hydrogen production vs imports
Overall, we find that green hydrogen imports appear in 2035 in the GH2E scenario with around 78 TWh, while reaching a peak of 182 TWh

in 2045 and then declining to 99 TWh by 2050. While the levelized cost of generating hydrogen in third nations (including both production and transportation costs to the geographical border of the model) is estimated to be lower than the average domestic European hydrogen production cost, the imported hydrogen is expected to make up ~17% of the anticipated import potential of 590 TWh by 2050 (Fig. 2b). This result highlights the cost competitiveness of domestic European hydrogen production compared to imports when considering endogenous expenses for investing in and transporting hydrogen to the regional demand centers. Furthermore, domestic hydrogen generation and low-cost underground storage offer flexibility to the European energy system.

Importing hydrogen from third nations impacts hydrogen production, where the electrolysis capacity increases from 466 GW to 518 GW by 2050 in the self-sufficient scenario compared to the GH2E scenario. Furthermore, modifications to the architecture

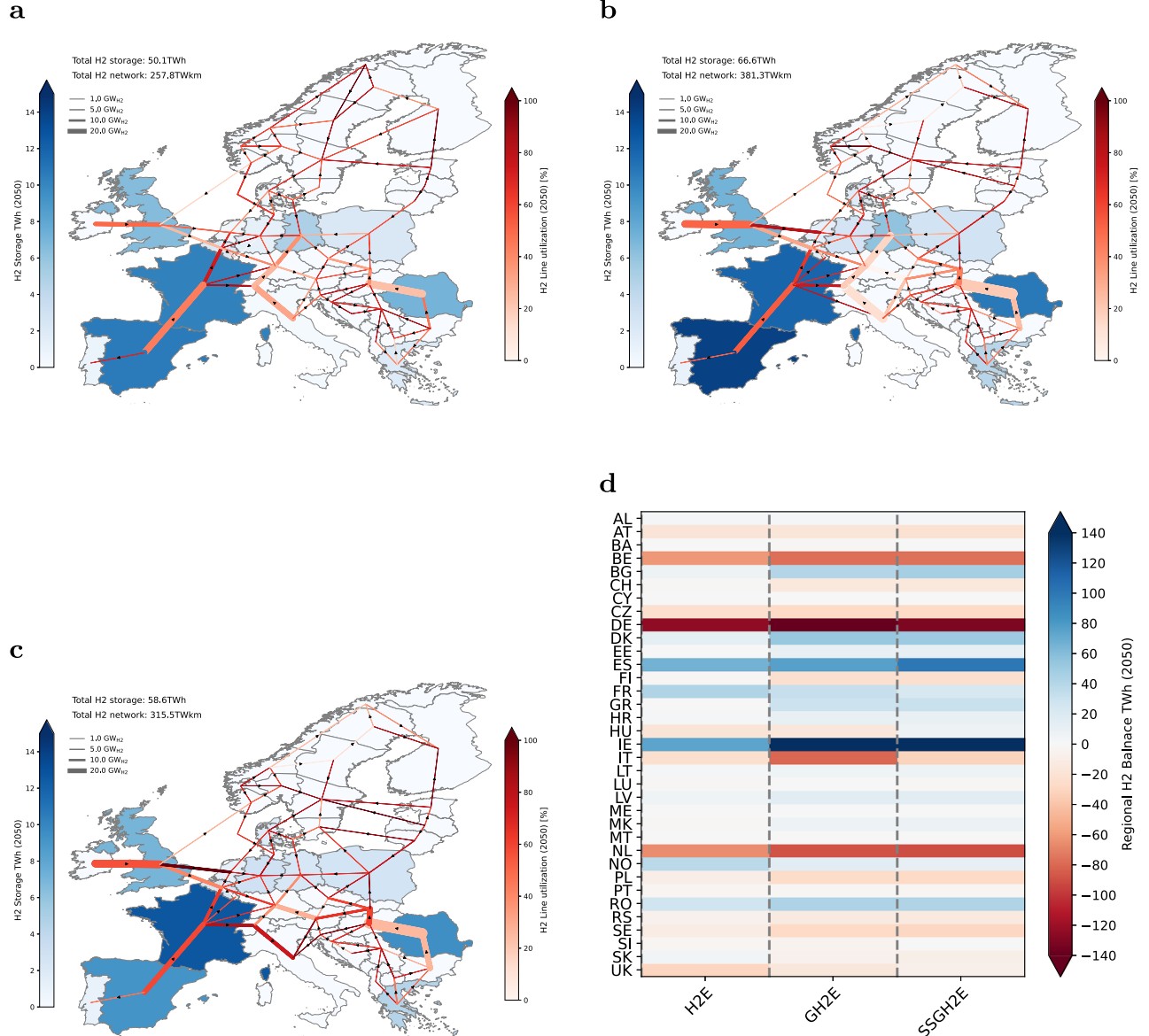

**Fig. 4 | Optimized European hydrogen infrastructure and hydrogen trading by 2050 for three main scenarios. a** Hydrogen Europe (H2E) (**b**) Green Hydrogen Europe (GH2E) (**c**) Self Sufficient Green Hydrogen Europe (SSGH2E). Optimal hydrogen grid and storage development for the three scenarios. The hydrogen infrastructure pathways from 2030 to 2045 can be found in Supplementary Figs. The line width depicts the hydrogen pipeline capacities of gigawatt $H_2$ ($GW_{H2}$) units. The white or darker shade red color shows the network's yearly utilization. The blue color tone represents the optimal installed hydrogen storage capacities in terawatt-hours (TWh) per region. The total network expansion is presented in terawatt-kilometers (TWkm) units. The arrow depicts the net flow volume direction. **d** Net hydrogen balances at a country level (i.e., production minus consumption) for the three scenarios, with dark blue signifying main hydrogen exporters and dark red suggesting importers. Country names are provided with ISO 3166-1 Alpha-2 code abbreviation[79].

of the hydrogen grid between scenarios with and without hydrogen import can be observed in Fig. 3c and d, where the Italian corridor is more evident by 2050 when allowing import from North Africa.

### Hydrogen infrastructure and cross-regional trading

A cross-regional hydrogen interconnection network offers trade of hydrogen between production and demand centers. We provide potential for repurposing the existing natural gas grid to hydrogen, which yields lower investment infrastructure costs (see Supplementary Information Fig. 3a and 3b) and thereby impacts investment decisions toward shaping the future hydrogen backbone network. Our results show a limited build-out of hydrogen infrastructure appearing in 2030, and we identify distinct hydrogen corridors connecting the outskirts

of Europe to the center in all scenarios by 2050, as visualized in Fig. 4. However, the size and utilization of cross-region hydrogen transmission lines can differ.

We find that cross-regional hydrogen interconnectors between countries where there exist opportunities to invest in seasonal storage facilities are designed with smaller capacities but have higher utilization due to the local balancing provided by the storage. This is exemplified by comparing the capacities of the Iberian peninsula and Italian corridors and utilization (Fig. 4), where the development of the Italian corridor is shaped by the lack of underground storage and the injected imports of hydrogen. We, therefore, discover that underground storage (-2.2 TWh total volume by 2030 and 50 TWh by 2050 in the H2E scenario) significantly impacts the design of a future European hydrogen infrastructure.

Furthermore, we observe significant changes both in infrastructure build-out and utilization when comparing scenarios with and without blue hydrogen. In scenarios without blue hydrogen, we find substantial and rapid growth in infrastructure build-out as well as long-term seasonal storage appear earlier to offset the highly volatile production of green hydrogen. Examples of modifications of the hydrogen infrastructure design are Ireland through the United Kingdom and Italian corridors, which are being expanded earlier by 2035 (see Supplementary Fig. 12b) due to increased hydrogen imports in the GH2E scenario and favorable onshore wind resources. Nonetheless, a sensitivity analysis regarding prospective imports of hydrogen derivatives from overseas reveals a reduction in the deployment of hydrogen networks and underground storage. For example, in the H2E, a potential scenario involving overseas derivative imports of 500 TWh by 2050 could lead to a 42% reduction in the hydrogen network's relative expansion, and in the GH2E, a 30% decrease in underground storage (see Supplementary Note 4).

Comparing the scenarios with and without imports of hydrogen, we find that a system without imports by 2050 (SSGH2E, Fig. 4c), could result in reduced transmission capacity size of 17.2% compared to the GH2E scenario (Fig. 4b) due to the expanded deployment of domestic hydrogen production.

Based on the previous findings regarding infrastructure development, the surplus of hydrogen production is most prevalent in the solar-rich southern regions and wind-rich coastal regions of Europe. Figure 4d displays the net difference in production and demand across the three scenarios. The domestic net volume (production minus consumption) of hydrogen traded in 2050 under the H2E, GH2E, and SSH2E scenarios is 414 TWh, 596 TWh, and 527 TWh, respectively. Independent of scenarios, large central-western European consumption centers demand ~50% of the traded hydrogen volume.

Another mechanism for reducing or altering the topology of future hydrogen networks could be the co-location of hydrogen production and demand for derivative fuels. We demonstrate that network expansion could be reduced by 18% in 2050 if the demand for 613 TWh hydrogen derivatives is not spatially fixed in the GH2E scenario. Furthermore, the new network development reduces the volume of imported hydrogen to countries like Germany, the Netherlands, and Belgium, decreasing from 300 TWh in the GH2E scenario

to 141 TWh with 60% spatial demand flexibility. Our analysis challenges the design of the envisioned European Hydrogen Backbone, which might be revised in light of spatial uncertainty in the demand for hydrogen derivatives (see Supplementary Note 5).

## Deployment of variable renewable energy generation

The electricity mix toward 2030 across the scenarios follows the same pathway with minor differences primarily due to the scenario design and relatively low hydrogen demand. However, toward 2050, the scenarios illustrating the plausible energy futures differ substantially, as do the electricity mix to supply the higher hydrogen demand and increased electrification within the energy sectors. We identify notable differences in final electricity consumption across the H2E and GH2E scenarios. By 2040, the absence of blue hydrogen necessitates an increase in electricity generation of 865 TWh (Supplementary Fig. 7). The H2E scenario suggests that the overall installed capacity for solar PV amounts to 2284 GW, while the onshore wind and offshore wind are 570 GW and 150 GW, respectively. Those investments for the GH2E are expanded by 17% for solar PV, 1% for onshore wind, and 39% for offshore wind. This illustrates that including blue hydrogen in the energy transition could reduce the deployment of additional renewable assets, particularly solar PV (see more in Supplementary Discussion 1).

Moreover, in the H2E scenario, the mean annual expansion rate of solar photovoltaic systems is 55 GW a$^{-1}$ for 2030 and 127 GW a$^{-1}$ for 2050, across the modeled countries. The GH2E scenario exhibits increased rates for solar PV, with 74 GW a$^{-1}$ by 2030 and 96 GW a$^{-1}$ by 2050. Comparable patterns are evident in the deployment of offshore wind energy. Furthermore, compared to the H2E scenario, in both GH2E and SSH2E scenarios, the additional production of green hydrogen tends to be located in regions with unique offshore wind potentials, such as Denmark, the Netherlands, and Ireland (see Fig. 5).

## Evaluating system costs of hydrogen pathways

Exploring three plausible, yet different, energy transition pathways by 2050, we find 2.77% higher system costs for a system solely depending on green hydrogen and imports (GH2E scenario), compared to a system where both blue hydrogen and import are possible options (H2E scenario). The additional annualized costs for the GH2E scenario are

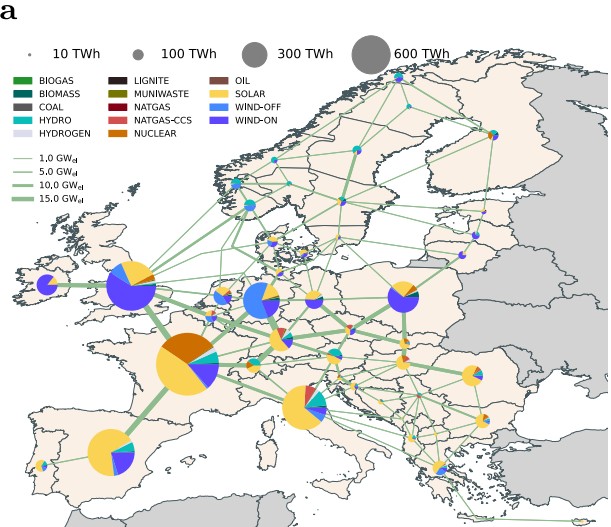

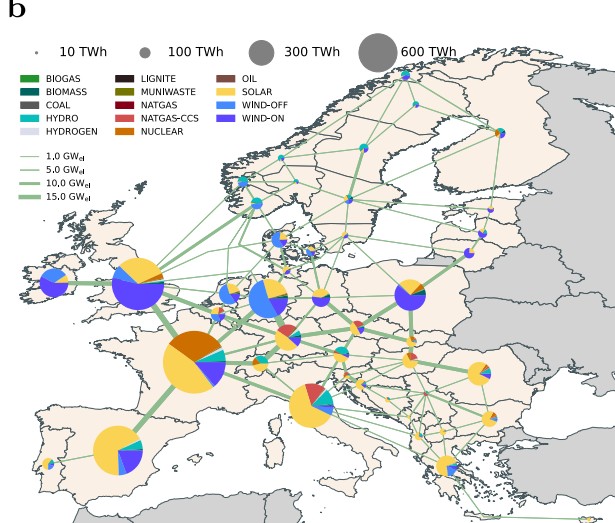

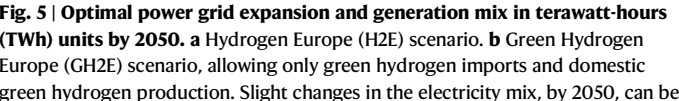

**Fig. 5 | Optimal power grid expansion and generation mix in terawatt-hours (TWh) units by 2050. a** Hydrogen Europe (H2E) scenario. **b** Green Hydrogen Europe (GH2E) scenario, allowing only green hydrogen imports and domestic green hydrogen production. Slight changes in the electricity mix, by 2050, can be noticed at the regional level between scenarios GH2E and Self Sufficient Green Hydrogen (SSGH2E) scenario. The pie charts show the percentage of technology utilized in each model region. The line widths depict the optimal installed capacity in gigawatt (GW$_{el}$) units of the electrical grid.

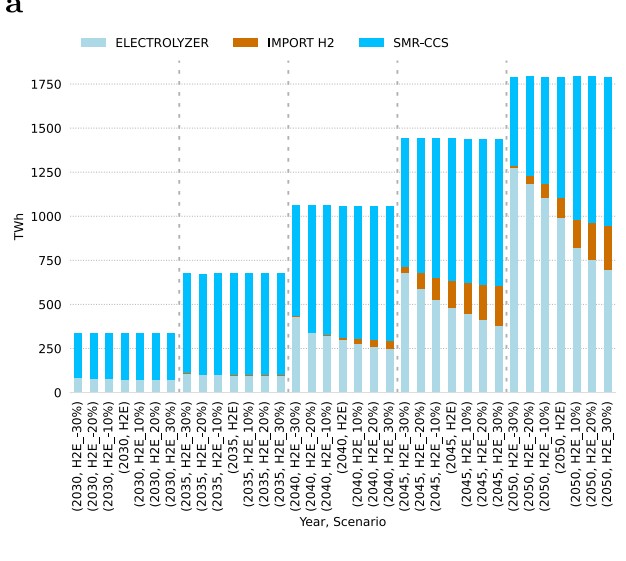

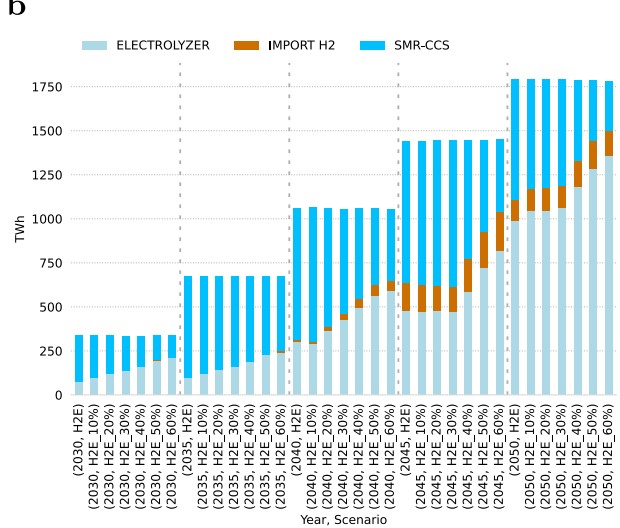

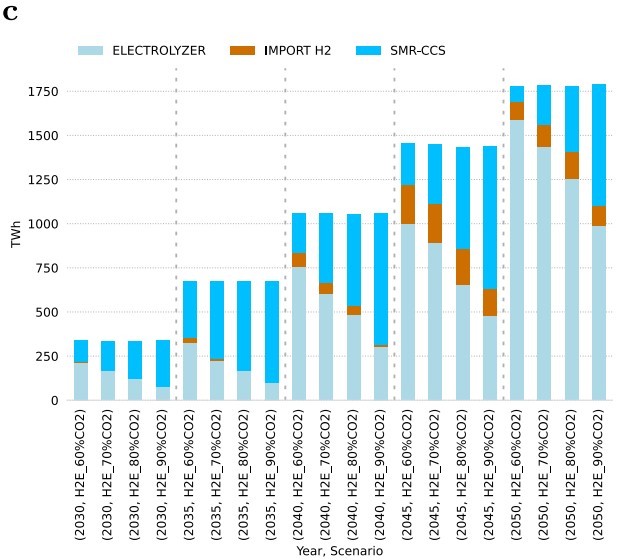

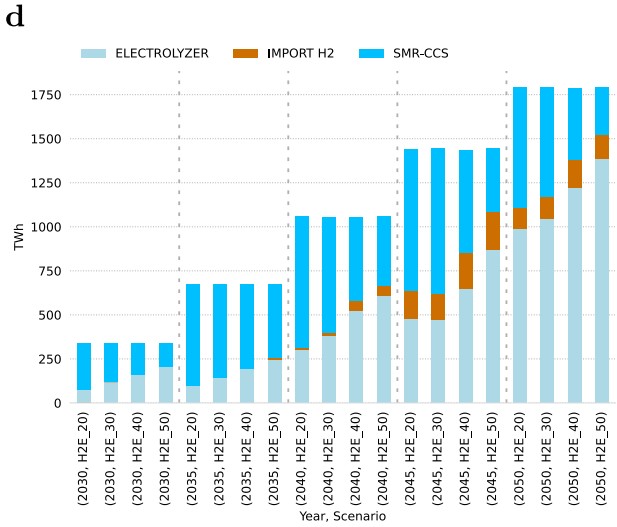

**Fig. 6 | Sensitivity analysis of hydrogen production pathways for the Hydrogen Europe (H2E) scenario.** Hydrogen production units in terawatt-hours (TWh). Main candidate technologies and methods: Electrolysis, Steam Methane Reforming with Carbon Capture and Storage (SMR-CCS), and hydrogen imports from neighboring countries via dedicated pipelines. **a** Varying capital expenditures (CAPEX) assumptions for water electrolysis. CAPEX discrete values range from − 30% to +30%. **b** Varying the natural gas price trajectory. Fuel price data are extracted by the World Energy Outlook (WOE 2022 - Net Zero Emissions scenario). Discrete values of the sensitivity range from +10% to +60%. **c** Varying the $CO_2$ capture rate by discrete values of 60%, 70% and 90%. **d** Varying the $CO_2$ cost € $tCO_2^{-1}$ for storing and transporting by 20, 30, 40, and 50.

therefore estimated to be around 14 billion € $a^{-1}$. Simultaneously, the self-sufficient green hydrogen scenario (SSH2E) is projected to have 3.36% increased costs compared to H2E, resulting in increased annualized costs of 17 billion € $a^{-1}$.

## Impact of tech costs and market dynamics

We evaluate the sensitivity of the H2E scenario's results by adjusting critical parameters of the supply-side hydrogen production technologies. Recent research[30] based on market surveys for alkaline electrolyzer technology reveals a significant spread between projected capital expenditures in 2030 and 2040. However, according to international organizations, there is widespread agreement that the cost of producing renewable hydrogen will continue to fall toward 2050[31,32] due to technological advancements and learning by doing[33]. Yet there are still lingering concerns on this assumption such as short-term scarcity of material and infrastructural availability[34]. First, if the water

electrolysis capital expenditures (CAPEX) decline by 30% in 2050 compared to the original projection (from 450 € $kW^{-1}$ to 315 € $kW^{-1}$), green hydrogen contributes to 71% (Fig. 6a) of the total hydrogen production. If the technological improvements and learning rates are not as high as expected, even with a 30% CAPEX increase, domestically produced renewable hydrogen penetration in 2050 remains above 40%. However, the European system would significantly rely on increased blue hydrogen production and extended hydrogen imports, accounting for -13.8% of the total production, or 247 TWh.

Although, traditionally, European natural gas market prices exhibit a seasonal and over the years fluctuating behavior[35], in recent years (2021–2023), analysts have expressed concern about their future evolution, which can impact the deployment of blue hydrogen. We find that a surge of up to 60% in the 2030 natural gas price (from 13.2 € $MWh^{-1}$ to 21.08 € $MWh^{-1}$), results in higher penetration of green hydrogen up to 63% (Fig. 6b). A similar increase by 2050 (from 10.9 €

MWh$^{-1}$ to 17.45 € MWh$^{-1}$) results in green hydrogen accounting for more than 76% of the total generation.

Carbon capture rate for CCS applications is estimated to vary from 56% to 90%[36]. With low capture rates, SMR technology as an industrial process becomes less competitive due to exposure to the rising European Union Emissions Trading System (EU ETS) price, referred to as $CO_2$ tax in this study. We discover that a low capture rate of 60%, compared with 90%, significantly impacts the competition between blue and green hydrogen by 2030, with green hydrogen penetrating at 64% and accounting for nearly 89% in 2050 (Fig. 6c). Finally, costs for pipeline transportation and $CO_2$ storage are yet uncertain and can influence the competition as well. A recent research study[37], for example, predicts a spread of € 3.6–41 t$CO_2^{-1}$. According to the sensitivity analysis, a cost of € 40 t$CO_2^{-1}$ results in an earlier expansion of green hydrogen with a total output of 160 TWh by 2030 and a subsequent increase to 1224 TWh by 2050 (Fig. 6d). A potential combination of high costs (€ 40 t$CO_2^{-1}$) and low capture rates (60%) results in a low blue hydrogen penetration in the short-term years 2030-2040, with blue hydrogen phasing out of the system by 2050 (see Supplementary Note 2). In the coming years, industrial and non-industrial CCS applications will compete for limited EU $CO_2$ storage projects[38]. If the expansion of operational storage facilities does not meet the demand between 2030 and 2040, the deployment of blue hydrogen could be decreased. Such a scenario would prioritize the development of green hydrogen and promote imports, demanding an earlier expansion of the hydrogen infrastructure as determined in the H2E scenario. For further analysis of limited $CO_2$ storage project development, see section 4.2 in the Supplementary Note 3.

## Discussion

Despite enthusiasm for the hydrogen economy, the establishment of an integrated European hydrogen infrastructure with enhanced sector coupling benefits remains a topic of ongoing discussion. Based on a large-scale energy system modeling analysis, we project the emergence of hydrogen production centers across Europe by 2030, with major centers likely located in the continent's periphery as we transition toward a low-carbon energy system by 2050. To meet the future estimated European hydrogen demand, we identify the development of four distinct hydrogen corridors from 1) Spain and France, 2) Ireland and the United Kingdom, 3) Italy, and 4) Southeastern Europe, facilitating the transportation of large volumes of hydrogen.

Our sector-coupled modeling approach contributes to the existing literature on the European energy transition and decarbonization, providing a long-term perspective on how hydrogen supply alternatives can diversify the future European hydrogen economy and infrastructure. In summary, we show that facilitating the energy transition requires rapid scale-up of electrolysis capacity, build-out hydrogen pipelines and storage facilities, deployment of renewable electricity generation technologies, and coherent coordination across European borders. However, in the event that hydrogen derivative demand is met through overseas imports, reaching 500 TWh by 2050, cross-border hydrogen connections could potentially be reduced by 25% and underground storage requirements by 30%. Notably, the 2030 estimated terminal capacity for hydrogen derivatives within the EU is ~146 TWh a$^{-1}$ or 4.4 Mt a$^{-1}$ [39]. In contrast, Re-PowerEU[3] anticipates hydrogen and derivatives imports of 10 Mt a$^{-1}$ by 2030, indicating that expanding port infrastructure is necessary to offset potential savings from a reduced hydrogen network and storage development.

Moreover, we highlight that the potential relocation of hydrogen demand for derivatives located mainly in industrial regions of Germany, the Netherlands, and Belgium towards European countries with competitive hydrogen production can impact the future topology and development of the hydrogen backbone.

While neighboring European nations proclaim ambitious hydrogen export goals through dedicated pipelines, we demonstrate that by 2030, domestic hydrogen production will be sufficient to meet the anticipated European demand. We show that European hydrogen production can be competitive when sector coupling synergies are considered towards 2050.

In contrast with other studies performing overnight investments by examining only 2050, a brownfield approach is adopted. Our analysis reveals a potential lock-in effect of blue hydrogen, based on natural gas, by the early 2030s when considering the competition of all major production technologies. However, this competition is dynamic. We demonstrate that potential technological changes, such as reduced capital expenditures for electrolysis, possibly inadequate carbon capture rates for CCS, limited $CO_2$ storage projects, and rising costs associated with the storage and transportation of $CO_2$ or future higher natural gas market prices, could tilt the favor towards green hydrogen. Europe can avoid this lock-in effect by exclusively utilizing green hydrogen from domestic electrolysis and green imports. Achieving a green hydrogen transition requires rapid expansion of the hydrogen networks by 2040, additional investments in renewable energy generation up to 865 TWh, and potentially increased reliance on green hydrogen imports from third nations. All at the expense of increased annualized system costs of around 2–3%. Furthermore, we highlight the importance of investing in large-scale underground storage of up to 66 TWh by 2050, particularly in salt caverns, in conjunction with the development of the European hydrogen backbone.

Potential barriers, such as short-term supply shortages caused by the momentum of project announcements or long-term uncertainty, could hinder water electrolysis deployment and should be addressed by appropriate regulation[34]. Furthermore, satisfying the future estimated European hydrogen demand requires a rapid scale-up of electrolysis capacity, where we find onshore wind and solar-rich regions particularly promising, followed by regions with unique offshore wind conditions. While our findings illustrate plausible futures, they do not consider principles like first-mover effects, learning-by-doing effects, differentiated financial conditions, and risks that might impact the results. In Supplementary Note 1, we provide a comprehensive discussion of the limitations associated with this research, and we compare our findings with other studies.

Establishing a hydrogen economy requires a balanced approach to short and long-term production and infrastructure planning. We demonstrate that by 2050, the future hydrogen network is responsible for transporting more than 50% of the traded hydrogen volume to demand centers in central-western Europe. Policy mechanisms should be implemented to mitigate disparities and ensure the equitable distribution of benefits from the future hydrogen backbone.

## Methods
### Balmorel energy system optimization model
Balmorel is an open-source[40], deterministic, partial equilibrium model for optimizing an energy system assuming perfect markets and economic rationality[27]. Similarly to other energy system models, it builds on a bottom-up approach and computes the least-cost solution for the energy system to satisfy various energy demands.

Balmorel is a technology-rich energy system model with a comprehensive representation of energy technologies and infrastructures. Energy sources are converted to energy vectors, which through transmission can be used to satisfy demands or be used by conversion technologies in different energy sectors. In parallel, it optimizes both investment planning and operational dispatch.

The modeling framework has been developed extensively by an open-source community since its first release in 2001[40]. The mathematical formulation and results have recently been compared against four other well-known open-source energy system models[41,42], with conclusions emphasizing the model's validity.

The model has been used to evaluate various energy transition scenarios throughout the years and is being developed to enable

holistic energy system evaluations. It was also used to perform deep-dive investigations of certain components of the energy system on various geographical scales and scopes. For example, Balmorel was applied to analyze the role of hydrogen in the future North European power system in 2060[43]. It was applied to provide decarbonization pathways for the Northern European integrated power and district heating system[44] with a focus on the role of renewable gas. Recently, it was utilized for assessing the future opportunities for offshore hydrogen production in Northern Europe[19] or diving deeper into the production of renewable transport fuels, including Power-to-X (PtX) and sector coupling opportunities[45,46].

## Balmorel modeling and data advancements
In comparison to previous model versions, the one used in this study demonstrates new developments and notable improvements. We expanded considerably the geographical coverage of the model to include the EU 27, the United Kingdom, Norway, Switzerland, and the remaining Balkan nations. The model encompasses a representation of all major energy sectors (see Fig. 1) and allows comprehensive sector coupling to investigate potential synergies between energy vectors and sectors. Furthermore, we improved the coverage of hydrogen-related elements such as geospatial allocation of prospective European hydrogen demands, hydrogen network development consisting of repurposed natural gas or new pipelines, adequate modeling of hydrogen underground storage in salt caverns, and interaction with third nations for trading hydrogen. In addition, we validate the technical renewable energy investment potentials (Supplementary Method 9) across multiple resources[23,47,48].

We build upon previous modeling advancements to update the model's geographic coverage. The power sector coverage (Supplementary Method 8) is expanded, and the modeling of electricity flow between regions utilizes the techniques (i.e., net transfer capacity) defined in ref. 49, assuming similar capital expenditures[50] (see Supplementary Fig. 4a and 4b). Data related to final electricity consumption are extracted from Eurostat[51]. Existing electricity interconnection data and prospective plans are consistent with the most recent pan-European electricity infrastructure development plan (TYNDP 2022)[52].

The heating sector is divided into individual users (residential and tertiary sectors), process heating (low, medium, and high-temperature) in the industry[53], and district heating[54]. To expand the geographical coverage of the model, relevant data for district heating and individual consumers are based on the most recent report on renewable space heating under the revised renewable energy directive[55]. Additionally, the industrial heat consumption by country is updated according to ref. 56.

Furthermore, we revise the future hydrogen demand per country (Supplementary Method 10) in accordance with the European Hydrogen backbone report[5]. A comparison of European hydrogen demand projections across multiple studies for 2030 and 2050 can be found in the Supplementary Method 11. We downscale industrial and transport country-level hydrogen demand projections to the geographical granularity of Balmorel by utilizing geographical information mapping of European industrial[57] and long-haul truck activities[58] (see Supplementary Fig. 8a and 8b). The rest of transport activities, such as buses and coaches, passenger cars, light commercial vehicles, and rails, towards 2050 are assumed to be decarbonized through direct electrification. Country-level demand projections for electrifying the transport sector are extracted from the EU Reference Scenario 2020[59].

The main hydrogen-related mathematical modeling is available in Supplementary Method 1. To evaluate the effect of synthetic fuel exogenous demands on the optimal sizing and hydrogen pipeline network topology, a spatial demand shift module is developed. New decision variables (see Supplementary Method 2) allow for exogenous

assigned synthetic fuel demand (hydrogen derivatives) to endogenously shift spatially to other model regions. Furthermore, a myopic modeling approach is also used due to the high complexity of the optimization problem. The differences between myopic, limited, and perfect foresight modeling methodologies are examined to assess the blue hydrogen lock-in effect. There are minor differences in the results (see Supplementary Method 3).

## Modeling hydrogen production and sector coupling
Hydrogen can be used for various purposes, e.g., 1) directly in the industrial sector for providing high-value heat or transport sector, or as peak power production, 2) to produce liquid PtX fuels, or 3) to produce synthetic methane, which can substitute natural gas. In Balmorel, demands for direct hydrogen in the industrial and transport sectors and liquid PtX fuels are defined exogenously. The final exogenous demand ranges from 326 TWh in 2030 to 931 TWh in 2040 to 1530 TWh in 2050. Furthermore, the need to use hydrogen for peak power production is endogenously calculated.

Hydrogen can be produced using different pathways, with the most prominent being 1) via alkaline water electrolysis, 2) using steam methane reforming (SMR) (gray hydrogen) and 3) using SMR with CCS (blue hydrogen). From the production facilities, hydrogen can be stored and transported via transmission infrastructure to its point of use. Currently, a hydrogen transmission infrastructure does not exist, but Balmorel is allowed to invest in new hydrogen infrastructure, such as hydrogen pipelines and storage facilities.

Furthermore, cross-sectoral synergies are incorporated into the modeling framework, e.g., by enabling excess heat from electrolytic hydrogen production to efficiently supply heating demands through district heating. The electrolyzer fleet can provide flexibility to the power system, and the operation is optimized endogenously in Balmorel. The downstream PtX production is less flexible, supplying a more stable demand. Thus the flexibility of the electrolyzer is subject to investment in and operation of storage facilities. Further details for hydrogen mathematical modeling description can be found in Supplementary Method 1.

## Carbon capture and storage
Carbon capture and storage (CCS) is a possibility for new investments in generation technologies. Due to the model complexity and the focus of the current study, we permit investments for CCS in technologies generating electricity, hydrogen, and heat. Due to the large influence of economies of scale for CCS, the CCS is allowed only for large-scale CHP and non-CHP plants such as (steam turbines, gas turbines, combined cycle, or engines)[54]. The $CO_2$ management is developed to account for transportation and storage costs (€ 20 $tCO_2^{-1}$)[60], similar average cost is provided by ref. 37. The capture rate is assumed to be 90%. A sensitivity analysis on the cost of transportation and storage as well as on the capture rate and limit on build-out rates of carbon storage potential is conducted (see Results section). Additional electricity consumption is accounted for in the capturing process (371 MWh $tCO_2^{-1}$ captured[61]), which reduces the net efficiency of the unit. Furthermore, we evaluate the technically accessible $CO_2$ storage resources based on the European Commission project CO2StoP[62]. The reanalysis results provide probabilistic estimates of resources for underground storage (saline aquifers and hydrocarbon fields). The European Commission recently published the Net Zero Industry Act[63] highlighting that a key bottleneck for the carbon capture investments is the lack of operating $CO_2$ storage sites. The European Commission sets a Union target of 50 Mt of annual operational $CO_2$ injection capacity by 2030 with a potential estimate of 550 Mt by 2050[63]. Our second scenario, GH2E, is motivated by the uncertainty in CCS deployment and the likelihood of extended natural gas consumption for low-carbon hydrogen production, blue hydrogen.

## Hydrogen infrastructure expansion network and storage

Hydrogen transport follows the same level of geographical aggregation as the electricity network and is modeled with transmission pipelines assuming linear bi-directional flow. In Supplementary Method 6, we include a cost comparison of hydrogen transport methods, highlighting pipelines as the most competitive option for European cross-border volumes trading. Based on pipeline size, the specific investment capital expenditures for hydrogen transmission pipelines and compressors are derived from the most recent European Hydrogen Backbone (EHB) report[8]. In accordance with the EHB report, pipeline investment expenditures are classified as either repurposed or new. For both types, the distance in a straight line between the centers of the modeled regions is estimated. Later, a weighted investment cost (€ MW$^{-1}$) per pipeline is computed based on the characteristics of the required infrastructure, such as onshore, offshore, new, or repurposed. Moreover, due to the relatively low demand for hydrogen in Europe, we assume that only medium-sized lines will be repurposed or newly invested until 2030. Meanwhile, economies of scale and learning rates for large cross-border pipelines can be accounted for beginning in 2040, lowering the expected capital expenditures (see Supplementary Table 1). In addition, costs and assumptions are made for the energy required to compress the hydrogen produced by water electrolysis, the expected pipeline lifetime, and hydrogen transmission energy losses for further information, see Supplementary Method 5.

Although the EHB characterizes which lines are classified as repurposed (first type), it does not provide information regarding their existing capacity. Therefore, we utilized the geographical information mapping of the existing methane European grid based on the SciGRID project[64]. We note that due to the small existing capacity, repurposing existing methane pipelines may still necessitate the construction of new hydrogen transmission in a few instances (e.g., cross-border connections between Spain and France). The proportion corresponding to the repurposed length is adjusted to reflect these specifics. In addition, the length split into offshore and onshore pipeline distance is determined based on the EHB reports. The investment costs of new pipelines (second type) are calculated using a similar methodology and breakdown costs. The final computed costs per pipeline can be seen in Supplementary Fig. 3a and Fig. 3b.

In this study, hydrogen can be stored in steel tanks or underground salt nearshore and onshore caverns[65]. While the hydrogen steel storage could, for the sake of simplicity, operate at the same pressure as the future hydrogen grid[19], the salt caverns' operational status could affect the internal gas pressure. To adequately capture the pressure differences when expanding hydrogen from caverns, we use the software REFPROP/NIST[66] to calculate the density of hydrogen at a given pressure and temperature. These parameters are incorporated into a simulation operational model (see Method 7, Supplementary Table 2) of 1 TWh of hydrogen underground storage to determine maximum discharge volumes per time period. For simplicity, the volume of the cavern is assumed as constant. We allow the cavern to operate between 180 and 105 bar at a constant temperature of 39 °C. A maximum drop of 10 bar is permitted due to concerns about geotechnical safety[65] limiting the maximum daily volume for discharge. The simulation tool provides the total amount of hours per charging or discharging cycle used later as input to Balmorel.

## Importing hydrogen from third nations

The Balmorel framework is expanded further to permit importing hydrogen flows from third-party nations outside the examined energy system borders. We notice two distinct modeling approaches. The first would require simulating the whole energy system of those countries, as well as the associated hydrogen transmission and transportation alternatives. The European modeling framework would incorporate additional investment expansion decisions. As a result, the problem's objective function will be revised to account for investment decisions of a larger and interconnected energy system. Yet, assuming perfect competition and rational decisions, this technique would lead to studying and addressing the question of the possible imported hydrogen volumes from other nations, realized by minimizing the total cost of an extended system not only limited to the European framework.

However, nations such as Algeria, Tunisia, Morocco, and Ukraine have already stated exporting ambitions and targets. With the second approach, we question whether those targets are competitive with domestic hydrogen production, and how imports would impact the development of the European future hydrogen infrastructure. We apply an external planning and operation optimization framework with the objective of minimizing system costs while meeting a yearly demand target for hydrogen generation. The optimization problem results in investments in technologies such as utility solar PV, onshore and offshore wind turbines, and hydrogen transmission pipes. The levelized cost of producing and transporting hydrogen via dedicated repurposed natural gas pipes to system boundaries is then estimated. The problem is addressed sequentially for the years of the announced targets (i.e., 2030, 2040, and 2050) from the three potential importing choices (i.e., Algeria and Tunisia, Morocco, and Ukraine). As exogenous input, the European modeling framework is updated with expected imported prices, pipeline capacities, and available yearly volumes. Those details act as the contact limits between the system boundaries and the third countries. More details regarding technological costs and outcome results can be found in Supplementary Method 12.

## Scenario choice and description

We conduct three modeling scenarios based on a least-cost optimization with a focus on examining the future hydrogen production pathways and infrastructure in Europe. The scenarios are based on the most promising and technologically feasible options for producing and storing hydrogen, as well as options for importing green hydrogen from other nations. The following sections provide extensive information for each scenario.

## Hydrogen Europe (H2E)

The hydrogen Europe (H2E) scenario addresses the study's main question on where, when, and how to produce hydrogen in a European energy setup. We want to offer insight into the competition between hydrogen-producing technologies in Europe, import possibilities, and information on potential hydrogen infrastructure. We allow the model to use all available technologies, including electrolysis (alkaline cells), conventional steam methane reforming (SMR), and steam methane reforming with CCS (SMR-CCS). Besides those production technologies, the model can invest in hydrogen storage, such as underground nearshore and onshore salt caverns or steel tanks. The subsurface formations are located in certain geographical areas and have a large potential for underground hydrogen storage[65]. In addition to internal European production, the H2E scenario also allows the importing of hydrogen through third-party nations (Morocco, Algeria and Tunisia, and Ukraine) based on the methodology described above. National electrolysis capacity targets up to 2030 are considered to depict a plausible short-term hydrogen market evolution[4].

Furthermore, over the long term, it is expected that the future hydrogen grid will complement the electricity grid reinforcements[13]. We pay specific attention to the electricity grid development in our scenario setting. According to ENTSO-E's Ten Year Network Development Plan (TYNDP) 2020, more than 300 transmission projects are expected to be completed by 2040[52]. Despite this, 60% of the projects are delayed or altered in some way[67]. Because those TYNDP projections are proving ambitious, we limit the electricity expansion grid to TYNDP across neighboring European countries until 2035. After 2035, the model co-optimizes power and hydrogen networks. The main model

input parameters for hydrogen grid infrastructure, technological investment costs (Supplementary Note 8), and assumptions are those discussed in Methods and in the Supplementary Information. This approach and restrictions are applied across all scenarios.

**Green H2 Europe (GH2E).** Fit for 55 packages[2] and the more recent RePowerEU[3] initiative aim at accelerating renewable hydrogen production while phasing out the dependency on fossil fuels. The latter plan necessitates a large European Electrolyzer capacity of around 64 GW by 2030[68]. The carbon tax pricing and fossil fuel price projections (Supplementary Note 7) in this study are based on the World Energy Outlook 2022 (Net Zero Emissions scenario, NZE)[69]. According to the NZE scenario, high $CO_2$ taxation assumptions ranging from 140 € ton$^{-1}$ in 2030 to 250 € ton$^{-1}$ in 2050 will result in lower demand for fossil fuels and, consequently, lower market prices. Therefore, low-carbon hydrogen produced from natural gas with CCS is projected to be economically viable. However, whether hydrogen produced from SMR-CCS can be considered low carbon is debated in the literature[36,70–73]. Some studies focus on methane leakages and life-cycle emissions, causing additional warming effects, whereas others assume higher capture rates, resulting in lower overall emissions. Another emerging challenge is that large-scale deployment of underground carbon storage leakage rates must be kept to less than 0.1% a$^{-1}$ on average, but methods for monitoring and confirming storage to this precision have yet to be established[74]. Furthermore, CCS, an immature technology with minimal public awareness, may face social acceptance challenges. However, there is evidence that it is possible to encourage social acceptance of CCS and perhaps avert demonstrations and opposition by presenting information on its environmental benefits[75]. Lastly, although blue hydrogen provides an alternative pathway[76], it conflicts with the European Commission's ambitions to accelerate the face out of natural gas and dependency on fossil fuels. These factors encourage us to explore the effects of a large-scale electrolysis expansion on the European energy system in the absence of blue hydrogen. We develop a policy scenario called Green H2 Europe (GH2E), in which we exclude the potential for investing in SMR-CCS starting in 2030. Although there is no direct restriction in the model that electrolytic hydrogen is produced with renewable electricity, with increasing $CO_2$ quota prices (Supplementary Note 6) assumed, the electricity production will increasingly become green, and so will the hydrogen. Yet, we continue to permit renewable hydrogen imports from third countries as RePowerEU proposes.

**Self Sufficient Green Hydrogen Europe (SSGH2E)**
Another contentious discussion is the risk of relying on the import of hydrogen from other countries[24,77,78]. In the final scenario, we subtract this opportunity. The model must determine the best approach to meet hydrogen demand using solely water electrolysis technology. Blue hydrogen investments are not permitted so as to decouple hydrogen production from conventional fuels for the concerns described in the GH2E scenario. This scenario strains the energy system and sheds light on European countries' competition for renewable energy resources for hydrogen generation while shaping an alternative hydrogen network without the effect of imports. Finally, a picture of a future European energy system that is self-sufficient in domestic green hydrogen generation is provided in this scenario.

## Data availability
The Raw input data for this study are openly available at Zenodo https://zenodo.org/records/10992469 under ISC license.

## Code availability
The latest Balmorel model development is openly available via Github https://github.com/balmorelcommunity/Balmorel under ISC license.

Furthermore, the source code developed for the current study can be accessed by the following GitHub branch.

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

## Acknowledgements

The authors would like to acknowledge financial support from the SuperP2G project that has received funding in the framework of the joint programming initiative ERA-Net Smart Energy Systems' focus initiative Integrated, Regional Energy Systems, with support from the European Union's Horizon 2020 research and innovation program under grant agreement No 775970. The opinions expressed in the manuscript are those of the authors and may not in any circumstances be regarded as stating an official position of the European Commission.

## Author contributions

I.K.: Conceptualization, Methodology, Software, Validation, Investigation, Data curation, Writing - original draft, Visualization. R.B.: Conceptualization, Methodology, Software, Supervision, Validation, Writing - original draft. T.M.: Methodology, Validation, Data curation, Writing review & editing. J.G.B.: Methodology, Software, Data curation, Writing review & editing. M. M.: Project administration, Supervision, Funding acquisition, Writing review & editing. D. K.: Conceptualization, Supervision, Validation, Funding acquisition, Writing review & editing.

## Competing interests

The authors declare no competing interests.
