## [Peer Review File · Nature Communications]

Reviewers' comments:

Reviewer #1 (Remarks to the Author):

The study explores potential pathways for an European hydrogen infrastructure using the fully sector-coupled energy system model Balmorel, identifying a potential medium-term lock-in effect of blue hydrogen and emphasizing the importance of rapidly scaling up electrolysis capacity, hydrogen network and storage facilities, renewable electricity generation, and coordination across European nations. The study also presents a scenario where a self-sufficient Europe relies on domestic green hydrogen by 2050, increasing yearly expenses by 3% and requiring 500 GW of electrolysis.

The main scientific novelty is the modeling the pathways of hydrogen corridors. Imports to Europe from different countries have been already explored in a sector coupled model (e.g. <https://doi.org/10.1016/j.renene.2021.08.016>, <https://doi.org/10.1016/j.renene.2023.04.015>) and hydrogen infrastructure with higher spatial resolution for a net-zero energy system has been already analysed in other publications (e.g. in <https://doi.org/10.1016/j.joule.2023.06.016>).

I have the following major concerns about the modeling assumptions which would need to be addressed for publication:

Major concerns:

1. Fixed spatial distribution of hydrogen demand and selection of entry points:

- The hydrogen demand seems to be largely fixed per region. This basically determines your hydrogen corridors. But e.g. H₂ demand, which is assumed for e.g. ammonia derivatives/synthetic fuels could move spatially. So they could be produced in other regions and then transported to the demand centers. This choice of either transporting the hydrogen to the demand centers or e.g. the synthetic fuel needs to be modeled or at least the impacts on the hydrogen corridors of e.g. removing the hydrogen demands for producing green ammonia and synthetic fuels explored. The EU plans to import also green ammonia and derivatives so the assumptions that only hydrogen and not derivatives can be imported and that all the hydrogen needs to be transported to the demand centers instead of e.g. the synthetic fuels do not seem to be profound.
- Regarding entry points into the European Energy System: You seem to consider only imports through pipelines but imports by terminals could completely change your hydrogen infrastructure. This needs to be added to the model to discuss hydrogen corridors.

- Imports of derivatives instead of hydrogen. This should be discussed and explored at least in a sensitivity analysis. This reduces the need for a hydrogen network/ hydrogen storage.

- Imports from other countries, this should be at least discussed.

2. Comparison of blue and green hydrogen

- It is unclear from the description how the CO₂ management is modeled. This has a major influence on the competitiveness of blue hydrogen and needs to be analysed, e.g. what is the assumed carbon capture rate of SMR+CC? What is your CO₂ storage potential? Do you model CO₂ transport and storing? Do you reach a net-zero system by 2050 in your results?

- Lock-in effect is probably overestimated due to myopic investment decisions. This needs to be discussed. How would the results change if the model had perfect foresight? There might be investments into blue hydrogen in 2030 just because the model does not have the foresight that CO₂ prices increase by 2035. But this is not how investments would be made.

Unclear modeling assumptions which needs to be clarified:

- What is the assumed carbon capture rate of blue hydrogen? How large is the CO₂ storage potential? Is CO₂ transport modeled or at least costs for the transportation/ storing of the CO₂ added?

- Which data are you using for the hydrogen salt cavern potentials?

- H₂ demand per region and H₂ usage should be listed. Is the H₂ demand per region fixed? This is going to impact your results since e.g. derivatives/synthetic fuels could be produced at another. Share of H₂ demand for Ammonia, synthetic fuels, high value chemicals should be listed

- What is the share of the total system costs of the hydrogen network?

Further comments:

- Figure 1

- Regarding the transport sector -> what about internal combustion cars? Shouldn't be there an arrow from Oil->Transport?

- No option to convert methane to electricity?

- What is the meaning of the connection Residential <-> District heating?

◦ Is vehicle to grid considered?

- p.3 “To supply the estimated hydrogen demand, we find Europe’s electrolyzer capacity ranging from 24 GW to 68 GW by 2030, and 310 GW to 507 GW by 2050”

Please provide additional information, e.g. how much hydrogen is produced in TWh or Mt?

- p.3 “However, hydrogen demand centers in the industrial sector, for transportation, renewable gas, and liquid fuels, are often concentrated in locations with limited renewable resources”

Can you really claim this with the given spatial resolution? Many countries are modeled as one region. Please justify!

- p.5 Figure 3 It appears that the full hydrogen production switches by 2030 from gray to blue or green in your scenarios? This should be discussed in the text!

- p.5 “The current hydrogen export national strategies could cover approximately 20 % of the domestic hydrogen demand in 2030, assumed in this study. “

REPowerEU targets to import 50% of domestic demand, why did you assume such a low level of import volumes? Is the constraint of allowing 20% of domestic demand to be imported binding?

- p.5 “. We, therefore, discover that underground storage (approximately 2.2 TWh total volume by 2030 and 50 TWh by 2050 in the H2E scenario) significantly impacts the design of a future European hydrogen infrastructure.”

This is a bold statement. What if derivatives instead of H2 would be imported? What would be the role of a H2 storage in this case?

- p.6 Figure 4 Why is there such a large hydrogen storage in France? France does not have a large potential for cheap underground storage (see e.g. <https://doi.org/10.1016/j.ijhydene.2019.12.161>)

- caption Figure 5: “Optimal power grid expansion and generation mix.” Please clarify the size of power grid expansion compared to today’s value?

- p.8 “ natural gas price (from 13.2 e/MWh to 21.08 e/Mwh)”

that is still quite low, futures for natural gas in 2030 are currently at 50 Eur/MWh. This would impact the usage of blue hydrogen production and should be explored in a sensitivity analysis.

- p.8 “In summary, we show that facilitating the energy transition requires rapid scale-up of electrolysis capacity, build-out hydrogen pipelines and storage facilities, deployment of renewable electricity generation technologies, and a coherent coordination across European borders.”

This statement is not justified by the study, one would need to explore e.g. the impacts of not having any hydrogen network at all. How would your results change if derivatives are imported? Maybe the pipelines are not necessary in this case.

- p.9 “Furthermore, the need to use hydrogen for peak power production is endogenously calculated.”

Is this amount varying between the scenarios?

- p.10 “and hydrogen transmission energy losses [20].”

How large are the assumed losses? What do they include (slipping?, compressor energy needs)? Compressors within Europe are normally operated with electricity, how is that included into the modeling?

- SI, p.5 “s. However, only liquid ammonia and methanol are currently competitive for importing without the need for re-conversion “

But most of the hydrogen could be replaced with methanol, e.g. in shipping, for power balancing... At least, there should be a sensitivity analysis towards the impact on the H2 network if all green Ammonia demand is imported.

- SI Figure 3, Figure a and b use same color scale, legend is hiding a number (1e6)
- SI p.10 section 1.8 The section should discuss the topic of industry reallocation and possible local shifts in demand.
- SI p.11 “We observe a final demand for hydrogen to be approximately 332 TWh or 10 Mt by 2030 and 1,767 TWh or 53 Mt by 2050. “ Your assumed demand for 2030 is half of the planned demand in REPowerEU (importing 10 Mt H2 and producing 10 Mt H2 within Europe). This assumption needs to be justified.
- SI, p.11 “. The report provides country-level hydrogen demand and its penetration into different uses such as ammonia synthesis, liquid fuels and high-value chemicals, high-temperature industrial process heat, and iron ore reduction with direct use of hydrogen. “ This is one of the most critical assumptions in the publication. You assume a fixed hydrogen demand per region which has strong impacts on the resulting hydrogen infrastructure. But e.g. ammonia/high value chemical/ liquid fuels could be produced at other regions and transported then to the demand centers, so the assumption of having a fixed hydrogen demand per region is not correct.
- SI p.12 Figure 4, label “H₂ demand”

Reviewer #2 (Remarks to the Author):

Title: A unified European hydrogen infrastructure planning to support the rapid scale-up of hydrogen production

Abstract of this paper :

This study examined plausible pathways for hydrogen infrastructure and various methods of hydrogen supply using open-source code model. They identify the effect of blue hydrogen and danger of methane usage for hydrogen production. They also highlight development of electrolysis for self-sufficient hydrogen production.

Although the approach of the paper is interesting, it needs further improvement.

Revision for this paper :

1. The conclusion of the study seems obvious. Please highlight the novelty of the paper.
2. (Page 2) Please avoid lump sum references. All references should be explained with detailed and specific description.
3. (page 2) The model used in this study seems old dated. Because hydrogen industry has changed a lot in recent years. Please explain important update list after 2018.
4. (section 2.2) Although this study analyzed importance of blue hydrogen, supplementary material or explanation of blue hydrogen is insufficient.
5. (section 2.4) Hydrogen can be stored in various forms such as LOHC, liquid, and gas. Please consider various hydrogen storage form for hydrogen delivery and storage.
6. (section 2.4) Hydrogen can be transported by pipe and tube trailer depending on distance. Please consider the use of various transport method for developing hydrogen pathway.
7. (section 2.4) In this study, geographical characteristics seems the most important factor for developing hydrogen pathway. The level of technology and policy of countries should be considered for developing the optimal pathway.
8. (section 2.7) In case of Japan and south Korea, actually production price of hydrogen gas is continuously increasing. The government is paying subsidies to fall the hydrogen price for market. Please explain the reason for hydrogen price reduction.
9. (supplementary material) Although many countries proclaimed ambitious goals, achievement is unsatisfactory. To develop realistic hydrogen pathway, the model should consider real achievement rate of country's roadmap.
10. (section 3) Rapid scale-up of electrolysis capacity is important, but it needs enormous investment. The model should consider current development situation of each technology.

Detailed Responses to Reviewers:

We would like to thank the two reviewers for their constructive and useful feedback. These were very helpful, and we were able to address all of the comments and adapt the submission accordingly. We have copied the entire set of comments below, with our reply in blue and the included paragraphs in italics, which are also noted in the track-changes version of the main manuscript in red colour.

Reviewer #1 (Remarks to the Author):

The study explores potential pathways for an European hydrogen infrastructure using the fully sector-coupled energy system model Balmorel, identifying a potential medium-term lock-in effect of blue hydrogen and emphasizing the importance of rapidly scaling up electrolysis capacity, hydrogen network and storage facilities, renewable electricity generation, and coordination across European nations. The study also presents a scenario where a self-sufficient Europe relies on domestic green hydrogen by 2050, increasing yearly expenses by 3% and requiring 500 GW of electrolysis.

The main scientific novelty is the modeling the pathways of hydrogen corridors. Imports to Europe from different countries have been already explored in a sector coupled model (e.g. <https://doi.org/10.1016/j.renene.2021.08.016>, <https://doi.org/10.1016/j.renene.2023.04.015>) and hydrogen infrastructure with higher spatial resolution for a net-zero energy system has been already analysed in other publications (e.g. in <https://doi.org/10.1016/j.joule.2023.06.016>).

The authors would like to thank the reviewer for the well-structured feedback and recommendations on improving the quality of our manuscript. Thank you for reading the manuscript in such detail. Furthermore, we have updated our literature review, including the most recent research studies. Please look at Page 1, section 1, Main Text.

We have further updated our conclusions and discussion section, highlighting the novelty of our work.

1) Methodological contribution performing pathways instead of examining the energy system in the final year (greenfield approach). Otherwise, lock-in effects could not be assessed. Furthermore, we provide insights into how the future hydrogen infrastructure (network and storage) could look in different plausible scenarios.

Page 8, section 3, Main Text:

In contrast with other studies performing overnight investments by examining only 2050, a brownfield approach is adopted.

2) Methodological and conceptual contributions. We investigate the possibility of hydrogen derivatives demanding spatial relocation and its effects on future European hydrogen network topology and development.

Page 8, section 3, Main Text:

Moreover, we highlight that the potential relocation of demand for hydrogen derivatives located mainly in industrial regions of Germany, the Netherlands, and Belgium towards European countries with competitive hydrogen production can impact the future topology and development of the hydrogen backbone.

3) Extended sensitivity analysis on the potential technological changes and natural gas market price signals on the competition of blue, green, and hydrogen imports.

Page 9, section 3, Main Text:

However, this competition is dynamic. We demonstrate that potential technological changes, such as reduced capital expenditures for electrolysis, possibly inadequate carbon capture rates for CCS, and rising costs associated with the storage and transportation of CO₂ or future higher natural gas market prices, could tilt the favor towards green hydrogen.

4) We provide insights that when sector coupling benefits and synergies are accounted for, making European domestic hydrogen competitive to imports from neighbouring countries via dedicated pipelines. In addition, we investigate the effect on European production centers and infrastructure due to overseas hydrogen derivative imports as a sensitivity analysis on the main two scenarios H2E and GH2E.

Page 8, section 3, Main Text:

While neighbouring European nations proclaim ambitious hydrogen export goals through dedicated pipelines, we demonstrate that by 2030, domestic hydrogen production is sufficient to meet the anticipated European demand. We show that European hydrogen production can be competitive when sector coupling synergies are considered towards 2050.

Major Concerns

1. Fixed spatial distribution of hydrogen demand and selection of entry points:
◦ The hydrogen demand seems to be largely fixed per region. This basically determines your hydrogen corridors. But e.g. H₂ demand, which is assumed for e.g. ammonia derivatives/synthetic fuels could move spatially. So they could be produced in other regions and then transported to the demand centers. This choice of either transporting the hydrogen to the demand centers or e.g. the synthetic fuel needs to be modeled or at least the impacts on the hydrogen corridors of e.g. removing the hydrogen demands for producing green ammonia and synthetic fuels explored.

We appreciate that the reviewer emphasized the significance of fixed demand allocation, which can affect hydrogen corridors. To address the concern, we perform additional modeling efforts to assess alternative scenarios where the spatial production of hydrogen derivatives is endogenously optimized by the model. Additional equations permit the spatial shift and allocation of the synthetic fuel demand throughout the entire Pan-European model regions in order to meet an overall system synthetic fuel demand target rather than at the regional level. We use a scalar gamma [0,1] to account for the percentage of synthetic fuel demand, which could potentially shift from one region to others. Consequently, the demand for synthetic fuels which are not fixed at regional level can vary based on the parameter gamma, allowing assessment of the impact that spatially moving the demand around could have on the hydrogen infrastructure and corridors. Therefore, the regional balance equation is now decomposed and accounts for direct hydrogen demand (exogenously defined pr. region) and demand for synthetic fuels (optimized for pan-Europe). We run multiple sensitivity analyses so that up to 60% of the synthetic fuel demand can shift. This sensitivity reveals the optimal location for future demand centers with competitive

production potential to accommodate synthetic fuel demand and address the potential impacts on future hydrogen corridors. Furthermore, we chose to perform the sensitivity on both the (H2E) and (GH2E) scenarios to capture the effects of blue hydrogen.

The mathematical modeling can be found in the Supplementary Material (SM), section 1.2. The main results are presented in the SM, section 3.3. We have reflected on the additional results in the main manuscript.

Page:5, sections 2.4, Main Text:

Another mechanism for reducing or altering the topology of future hydrogen networks could be the co-location of hydrogen production and demand for derivative fuels. We demonstrate that network expansion could be reduced by 16% if the demand of 613TWh hydrogen derivatives is not spatially fixed in the GH2E scenario (see Supplementary Material, section 3).

Page:5, sections 3, Main Text:

Moreover, we highlight that the potential shifting of hydrogen derivatives demands located mainly in industrial regions of Germany, the Netherlands, and Belgium towards European countries with competitive hydrogen production can impact the future topology and development of the hydrogen backbone.

The EU plans to import also green ammonia and derivatives so the assumptions that only hydrogen and not derivatives can be imported and that all the hydrogen needs to be transported to the demand centers instead of e.g. the synthetic fuels do not seem to be profound.

- Regarding entry points into the European Energy System: You seem to consider only imports through pipelines but imports by terminals could completely change your hydrogen infrastructure. This needs to be added to the model to discuss hydrogen corridors.
- Imports of derivatives instead of hydrogen. This should be discussed and explored at least in a sensitivity analysis. This reduces the need for a hydrogen network/ hydrogen storage.

We address the concern about possible imports of derivatives and thereby also assume hydrogen demand levels by designing a sensitivity analysis where hydrogen derivatives are imported, thereby stepwise reducing the demands of derivative fuels to be produced at a European level. In this way, we address the impacts on production centers, possible corridors, and underground storage development.

The sensitivity analysis can be found in the SM, Section 3.2. We have reflected on the additional results in the main manuscript.

Page:5, sections 2.4, Main text:

Nonetheless, a sensitivity analysis regarding prospective imports of hydrogen derivatives from overseas reveals a reduction in the deployment of hydrogen network and underground storage. For example, in the H2E, a potential scenario involving overseas derivative imports of 500TWh by 2050 could lead to a 32% reduction in the hydrogen network's relative expansion (see Supplementary Material, section 3).

Page:8, section 3, Main Text:

However, in the event where hydrogen derivative demand is met through overseas imports, reaching 500 TWh by 2050, cross-border hydrogen connections could potentially be reduced by 16 % and underground storage requirements by 23 %. It is important to note, though, that the 2030 estimated terminal capacity within the EU is relatively low, approximately estimated as 146 TWh/a or 4.4 Mt/a [38].

In addition, we acknowledge that the location of terminals may alter the network topology. However, there is significant uncertainty about the future location of hydrogen derivative import terminals (see ref. [38], Report: "Facilitating hydrogen imports from non-EU countries", Guidehouse 2022).

Regarding the optimal transportation of hydrogen or derivatives, in SM section 1.6, we performed a literature review. We elaborated further on the cost differences and competitiveness between pipelines and other means for transporting hydrogen or derivatives. Recent studies show that hydrogen pipelines are the most competitive option over long distances compared to others, such as ammonia pipelines or ships, LOCH pipelines or rails, and hydrogen trucks in gas or liquid form. (Di Lullo et al. 2022 <https://doi.org/10.1016/j.ijhydene.2022.08.131>, report: "Global Hydrogen Trade to Meet the 1.5°C Climate Goal: Technology Review of Hydrogen Carriers" IRENA 2022).

Page:11, section 4.5, Main Text:

In the Supplementary Material (see section 1.6), we include a cost comparison of hydrogen transport methods, highlighting pipelines as the most competitive option for European cross-border large volumes trading.

Finally, we note that we have elaborated on the effects of imports in our self-sufficient scenario (SSGH2E). For example, see main text, sections 2.3 and 2.4.

◦ Imports from other countries, this should be at least discussed.

Based on the additional alternative sensitivity scenarios, we assess the impact on the importance of imports and discuss it further in the Main Text section 2.4 and SM sections 1.6 and 3.2.

2. Comparison of blue and green hydrogen

◦ It is unclear from the description how the CO₂ management is modeled. This has a major influence on the competitiveness of blue hydrogen and needs to be analysed, e.g. what is the assumed carbon capture rate of SMR+CC? What is your CO₂ storage potential? Do you model CO₂ transport and storing?

We thank the reviewer for raising concerns regarding CO₂ management. In the current manuscript, we have added a new subsection 4.4, under the methods, page 10. We further update the information regarding CO₂ management-related elements. We highlight the assumption behind the CO₂ storage potential, where we utilize the European Commission project CO₂StoP

(https://energy.ec.europa.eu/publications/assessment-co2-storage-potential-europe-co2stop_en, see Tables 2-11). Further details for transportation and storage costs are extracted from the Danish Energy Agency technology catalogues (see <https://ens.dk/en/our-services/projections-and-models/technology-data/technology-data-carbon-capture-transport-and>).

To address the reviewer's concerns regarding the effect of capture rates and the potential storage and transportation cost on the competition between blue and green hydrogen, we have assessed the impact by performing a sensitivity analysis. First, we perform a sensitivity analysis on the capture rates. Second, a sensitivity analysis of transportation and storing costs for CO₂. Lastly, we combined both and performed a sensitivity analysis based on the most pessimistic assumptions. The additional results can be found in the Main Text section 2.7 and SM section 3.1. The sensitivity analysis shows that the blue hydrogen lock-in effect exists, yet the final share of blue vs green may change based on data assumptions and projections.

Page:7, sections 2.7, Main Text:

Carbon capture rate for CCS applications has been estimated to vary from 56 % to 90 % [36]. With low capture rates, SMR technology as an industrial process becomes less competitive due to exposure to the rising EU ETS price, referred to as CO₂ tax in this study. We discover that a low capture rate of 60 %, compared with 90~%, significantly impacts the competition between blue and green hydrogen by 2030, with green hydrogen penetrating at 60 % and accounting for nearly 77 % in 2050 (Extended Data Fig. 6c). Finally, costs for pipeline transportation and CO₂ storage are yet uncertain and can influence the competition between blue and green hydrogen. A recent research study [37], for example, predicts a spread of 3.6 to 41 €₂₀₂₂/tCO₂. According to the sensitivity analysis, an uncertain cost (40 €₂₀₂₂/tCO₂) results in an earlier expansion of green hydrogen with a total output (160TWh by 2030) and a subsequent increase to 1224 TWh by 2050. Finally, a potential combination of high costs (40 €₂₀₂₂/tCO₂) and low capture rates (60 %) can lead to a moderate blue hydrogen penetration in the short-term years 2030-2040, with blue hydrogen facing out of the system by 2050 (see Supplementary Material, section 3).

Do you reach a net-zero system by 2050 in your results?

Regarding the net zero question. In this study, we apply a CO₂ cost. The final energy system, in the H2E, does not achieve a net-zero carbon footprint by 2050, but it comes very close. Emitting approximately 59.4 Mt of CO₂, which is 0.00247% of the 2022 emissions from fuel fossil fuel combustion for energy use in the EU territory (Eurostat 2022 <https://ec.europa.eu/eurostat/web/products-eurostat-news/w/ddn-20230609-2>, 2.4 Gt of CO₂ emissions).

◦ Lock-in effect is probably overestimated due to myopic investment decisions. This needs to be discussed. How would the results change if the model had perfect foresight? There might be investments into blue hydrogen in 2030 just because the model does not have the foresight that CO₂ prices increase by 2035. But this is not how investments would be made.

We acknowledge the reviewer's concern about myopic investment decisions. To address this concern, we perform additional model runs utilizing both myopic, limited foresight, and full perfect foresight runs to assess the impacts of model foresight on lock-in effects and investment decisions. Balmoral already encompasses the functionality of using different yearly foresight (myopic, limited foresight, or perfect foresight). However, the latter is extremely computationally heavy. We performed the assessment, which now brings a new layer to our analysis and enables more robust conclusions about the potential role and competition between blue and green hydrogen.

The analysis can be found in section 1.3 in the SM. Firstly, we acknowledge that the computational time increased dramatically when moving from myopic to perfect foresight. However, due to the ambitious CO₂ tax scenario, our results are found to be robust against the different foresight modeling techniques. Other studies, see Siala et al. (2022), with a European energy scope on foresight modeling demonstrate similar insights (<https://doi.org/10.1016/j.energy.2022.123301>).

Page 6, section 1.3, Supplementary Material:

Figure 1 depicts that independent of the chosen model foresight technique, the lock-in effect is projected and captured under the current data and system assumptions. Our findings are supported by the literature. A recent study by Lambert et al. (2023) [5] highlights that although the final European energy system looks almost the same across the two methodologies, there could be deviations in the intermediate years. However, another study by Siala et al. (2022) [6] compares optimal generation expansion of a European energy system results for five different models assuming either myopic or perfect foresight. Similar to our analysis, they demonstrate that under a high CO₂ price scenario, the result differences are not significant. Furthermore, we highlight that the computational time dramatically decreases when shifting from perfect to myopic foresight. For instance, the myopic solution required 22 hours of solution time. The perfect foresight resulted in a simulation run of more than 364 hours. Babrowski et al. (2014) [7] noted that the myopic approach with stable input parameters is just as applicable as the perfect foresight approach, with the added advantage of requiring significantly less computing time.

Page:9, sections 4.2, Main Text:

The differences between myopic, limited, and perfect foresight modeling methodologies are examined to assess the blue hydrogen lock-in effect. There are minor differences in the results (see Section 1.3 in the Supplementary Material).

Unclear modeling assumptions which needs to be clarified:

- What is the assumed carbon capture rate of blue hydrogen? How large is the CO₂ storage potential? Is CO₂ transport modeled or at least costs for the transportation/ storing of the CO₂ added?

Please see the previous comment where we addressed these concerns.

- Which data are you using for the hydrogen salt cavern potentials?

We have updated Section 4.5, Main Text. Based on the Caglayan et al. (2020), we use the onshore and near-onshore potentials.

Page 10, section 4.5, Main Text:

In this study, hydrogen can be stored in steel tanks or underground salt nearshore and onshore caverns [65]

- H2 demand per region and H2 usage should be listed. Is the H2 demand per region fixed? This is going to impact your results since e.g. derivatives/synthetic fuels could be produced at another. Share of H2 demand for Ammonia, synthetic fuels, high value chemicals should be listed.

Please, see the previous comment about addressing the concern about spatially distributed hydrogen derivatives demand and its effects in case its not fixed per region.

In addition, in section 1.10 of the supplementary materials, we have included maps illustrating the assumed regional demand allocation. The demand for hydrogen can be separated into endogenous and exogenous components. In the SM Section 1.10, additional information regarding the assumption for exogenous demand for transport, ammonia, and high-value chemicals, and Industrial heat and Steel is provided in tables and maps.

Page 15, section 1.10, Supplementary Material:

The final exogenous hydrogen demand allocation can be found in Table 4. The model endogenously optimized the direct hydrogen demand for peak power production. Table 3 illustrated the optimal results per scenario.

- What is the share of the total system costs of the hydrogen network?

Overall compared to the annual total expenses captured by the objective function, the hydrogen network's cost is low (see below table 1). However, the total share differs for every scenario. The shares support the main findings that the GH2E scenario requires earlier and more significant hydrogen network expansion investments in comparison with H2E. On the other hand, in the case of the Self-sufficiency scenario (SSGH2E), due to relatively increased domestic hydrogen production, the grid investments are reduced. A similar range of hydrogen network share costs is presented by Neumann et al. 2023, (<https://doi.org/10.1016/j.joule.2023.06.016>). In addition, we note our assumptions of a 4% discount rate and a lifetime expectancy of 50 years for network infrastructure investments.

% System cost	2030	2035	2040	2045	2050
H2E	0.03%	0.03%	0.06%	0.16%	0.30%
GH2E	0.10%	0.19%	0.26%	0.37%	0.49%
SSGH2E	0.10%	0.15%	0.23%	0.28%	0.40%

Table 1: Annual hydrogen network expansion costs share.

Further comments:

- Figure 1

- Regarding the transport sector -> what about internal combustion cars? Shouldn't be there an arrow from Oil->Transport?

We do not model the fossil fuel-driven transport means; rather, we model the changing electrification percentages based on the assumption that individual transport sector consumers will be electrified towards 2050, as well as increasing shares of hydrogen for synthetic fuels to decarbonize the long-haul transportation sector. For further information, see:

Main Text, page 9, section 4.2.

The rest of transport activities, such as buses and coaches, passenger cars, light commercial vehicles, and rails, towards 2050 are assumed to be decarbonized through direct electrification. Country-level demand projections for electrifying the transport sector are extracted from the EU Reference Scenario 2020 [60].

- No option to convert methane to electricity?

Thank you for notifying us of this. We model the option of converting methane to electricity and have updated *Figure 1. Page 2, Main Text*

- What is the meaning of the connection Residential <-> District heating?

Residential heating can be supplied either by individual technologies or by connecting to a district heating network. We elaborate on section 4.2, paragraph three. Furthermore, we provide additional details about the methods and data assumptions. We elaborated further by referring to the related publication.

Page 9, section 4.2, Main Text:

Further district heating modeling information and prospective expansion to unconnected individual users or low-temperature industrial process heat areas, as well as their associated technical expenses, can be found in ref. [54]

- Is vehicle to grid considered?

No, it's not in our model. Balmorel has the option to perform V2G. However, the add-on layer of information increases the computational burden heavily. We considered it out of the scope of this study.

- p.3 "To supply the estimated hydrogen demand, we find Europe's electrolyzer capacity ranging from 24 GW to 68 GW by 2030, and 310 GW to 507 GW by 2050"

Please provide additional information, e.g. how much hydrogen is produced in TWh or Mt?

We have updated the Main Text and provided the related information.

Page 3, section 2.1, Main Text:

To supply the estimated hydrogen demand, we find Europe's electrolyzer capacity ranging from 24 GW (73 TWh) to 68 GW (320 TWh) by 2030, and 310 GW (989 TWh) to 507 GW (1708 TWh) by 2050, depending on the scenario (see Fig. 3a, with values in parentheses denoting the corresponding production levels)

- p.3 “However, hydrogen demand centers in the industrial sector, for transportation, renewable gas, and liquid fuels, are often concentrated in locations with limited renewable resources” Can you really claim this with the given spatial resolution? Many countries are modeled as one region. Please justify!

We appreciate the reviewer's observation. We have rephrased it, and further clarification has been incorporated to enhance our statement. In addition, in the Supplementary Material, Section 1.10, we present the hydrogen demand per model region (see Table 4, page 17). Finally, related maps (i.e., Fig 6-8) illustrate spatially the final demand differences. Lastly, we note that the VRE resources are disaggregated into several categories (resource grades), which reduces the limitation of using the country as a node. For further information regarding Solar and Wind modeling, see Supplementary Material, Section 1.9.

Page 3, section 2.1, Main Text:

However, significant hydrogen demands for steel, process heat, transportation, ammonia, and high-value chemicals, are frequently concentrated in European regions/countries with lower renewable resources. For further information, see Supplementary Material, section 1.10.

- p.5 Figure 3 It appears that the full hydrogen production switches by 2030 from gray to blue or green in your scenarios? This should be discussed in the text!

We have elaborated further on the topic. Furthermore, we have added information in the Figure 3 caption.

Page 4, section 2.1, Main Text:

Our results shed light on the competition between gray, blue, and green hydrogen production pathways. Gray hydrogen is outcompeted based on a high CO₂ taxation projection (150 e/tCO₂, by 2030) implemented across the scenarios.

- p.5 “The current hydrogen export national strategies could cover approximately 20 % of the domestic hydrogen demand in 2030, assumed in this study. “

REPowerEU targets to import 50% of domestic demand, why did you assume such a low level of import volumes? Is the constraint of allowing 20% of domestic demand to be imported binding?

In this study we follow the assumptions of the European Hydrogen Backbone (EHB) reports. According to the report, the main hydrogen-importing nations are Algeria, Tunisia, Morocco, and Ukraine's with corresponding national targets. This analysis does not examine or follow the RePowerEU assumptions since they are considered among the most ambitious towards 2030. Please look at our new section 1.11 in the Supplementary Material, where we perform a review on the future European hydrogen demand scenarios (2030 and 2050) based on multiple references. We have added two new Figures 8a and 8b.

In addition, we emphasize that the model does not import hydrogen by 2030 across our scenarios. Our results show that imports have a moderate role toward 2050, highlighting the competitiveness of domestic hydrogen production when sector coupling benefits are accounted for. By 2050, in the H2E scenario, the total import level from all the neighbouring nations is 118TWh, and the potential is 590TWh based on the national strategies. The 20% refers to the optimal imported volume by 2050 (118/590). We apologize for the confusion. We have elaborated further in the main text and rephrased the statement:

Page 5, section 2.3, Main Text:

While the levelized cost of generating hydrogen in third nations (including both production and transportation costs to the geographical border of the model) is estimated to be lower than the average domestic European cost, the imported hydrogen is expected to make up approximately 20 % of the anticipated import potential of 590 TWh by 2050 (Fig. 3b).

• p.5 “ We, therefore, discover that underground storage (approximately 2.2 TWh total volume by 2030 and 50 TWh by 2050 in the H2E scenario) significantly impacts the design of a future European hydrogen infrastructure.”

This is a bold statement. What if derivatives instead of H2 would be imported? What would be the role of a H2 storage in this case?

Based on our additional alternative scenario analyses regarding the potential import of hydrogen derivatives, we show the potential reduction in hydrogen infrastructure requirements. The analysis is described in the Supplementary Material, section 3.2.

We have added additional information to the main text as follows:

Page 5, section 2.3, Main Text:

Nonetheless, a sensitivity analysis regarding prospective imports of hydrogen derivatives from overseas reveals a reduction in the deployment of hydrogen network and underground storage. For example, in the H2E, a potential scenario involving overseas derivative imports of 500TWh by 2050 could lead to a

32 % reduction in the hydrogen network's relative expansion a 23% decrease in underground storage (see Supplementary Material, section 3).

Page 33, section 3.2, Supplementary Material:

As Figures 12a - 12b show, substituting domestic ammonia and high-value chemicals hydrogen demand with overseas imports, decreases overall storage and network investment. By 2035 and 2040, in the H2E scenario, the effect of shipping imports is minimal due to the presence of blue hydrogen serving as a baseload production technology, satisfying the overall lower domestic hydrogen demand. On the contrary, in the GH2E scenario, shipping imports can have a significant impact on the requirement for European infrastructure development due to the intermediate production of green hydrogen. For example, in 2035, the storage and network expansion decreased by 38 % (Fig. 13a) and 50 % (Fig. 13b), respectively, for a 60 % shipping import scenario. Looking further into the future, in the 2050 H2E scenario, 84TWh shipping imports result in a 3 % network relative change, while 500TWh imports result in a 32 % (Fig. 12b). In the GH2E scenario, the network relative reduction is lower by approximately 16 %, and the underground storage is decreased by 23 %.

- p.6 Figure 4 Why is there such a large hydrogen storage in France? France does not have a large potential for cheap underground storage (see e.g. <https://doi.org/10.1016/j.ijhydene.2019.12.161>)

Our cavern potential is based on the same reference as pointed out by the reviewer, Caglayan et al. (2020). In that study, France has a potential of 500TWh of hydrogen stored in salt caverns Figure 8. At the same time, the model in our study invested less than 14TWh across multiple scenarios.

- caption Figure 5: “Optimal power grid expansion and generation mix.” Please clarify the size of power grid expansion compared to today's value?

Thanks also for this very relevant point. In section 4.6.1, we have provided a description of the H2E scenario, that we have prohibited optimal grid expansion up to TYNDP 2035 projections due to delays cited in [59]. After 2035, we permit the co-optimization of power and hydrogen networks, but only up to an additional 10GW of interconnector capacity. The same restrictions apply to the rest of the scenarios. We have elaborated further in the Supplementary Material:

Page 12, section 1.8, Supplementary Material:

Lastly, Fig. 5 illustrates the optimized network development. The network expansion is limited to TYNDP projections for 2035 until that year (see section 4.7, Main Text). The model then determines optimal capacities but is limited to an additional capacity of up to 10GW per cross-border line connection. In comparison to 2020, the results indicate that the electricity network could grow by a factor of four by 2050. Minor variations are noted among the scenarios.

- p.8 “ natural gas price (from 13.2 e/MWh to 21.08 e/Mwh)”
that is still quite low, futures for natural gas in 2030 are currently at 50 Eur/MWh. This would impact the usage of blue hydrogen production and should be explored in a sensitivity analysis.

Section 2.7 of the main text contains a comprehensive sensitivity analysis of natural gas price projections entitled "Future technological advancement and fuel price can impact the hydrogen production pathway." This analysis aims to examine the competitive dynamics among hydrogen imports, blue, and green. We acknowledge that EU natural gas market prices have increased by more than 50 €/MWh historically as a result of the last two years of the energy crisis; however, the World Energy Outlook 2023 price projection indicates that this increase is temporary. We emphasize that a recent study (Boitler et al., 2023) that examines decarbonization in the European Union and Fit for 55, projects that natural gas prices in the EU will fluctuate between 5.77€/GJ (20.77 €/MWh) and 6.76€/GJ (24.33 €/MWh) by 2030 (<https://doi.org/10.1016/j.joule.2023.11.002>). In light of the current natural gas price sensitivity analysis, we believe that an extreme natural gas price will demonstrate that green hydrogen will become even more competitive by 2030.

- p.8 "In summary, we show that facilitating the energy transition requires rapid scale-up of electrolysis capacity, build-out hydrogen pipelines and storage facilities, deployment of renewable electricity generation technologies, and a coherent coordination across European borders."

This statement is not justified by the study, one would need to explore e.g. the impacts of not having any hydrogen network at all. How would your results change if derivatives are imported? Maybe the pipelines are not necessary in this case.

We thank the reviewer for the careful consideration of our conclusions. We have reformulated the revised section by including additional insights and information. References on the possibility of overseas hydrogen derivative imports, relocation of demand have been added:

Page 8, section 3, Main Text:

However, in the event where hydrogen derivative demand is met through overseas imports, reaching 500 TWh by 2050, cross-border hydrogen connections could potentially be reduced by 16 % and underground storage requirements by 23 %. It is important to note, though, that the 2030 estimated terminal capacity within the EU is relatively low, approximately estimated as 146 TWh/a or 4.4 Mt/a [38]. Moreover, we highlight that the potential shifting of hydrogen derivatives demand located mainly in industrial regions of Germany, the Netherlands, and Belgium towards European countries with competitive hydrogen production can impact the future topology and development of the hydrogen backbone.

- p.9 "Furthermore, the need to use hydrogen for peak power production is endogenously calculated." Is this amount varying between the scenarios?

Thank you for bringing this point up. We have provided those numbers in the Supplementary material. The amount varies between the scenarios.

Page 16, section 1.10, Supplementary Material:

The model endogenously optimized the direct hydrogen demand for peak power production. Table 3 illustrates the results per scenario.

- p.10 “and hydrogen transmission energy losses [20].”

How large are the assumed losses? What do they include (slipping?, compressor energy needs)?

Compressors within Europe are normally operated with electricity, how is that included into the 14odeling?

We have further elaborated our techno-economic assumptions. We note that the electricity energy for compressor needs is accounted for implicitly as a percentage of the overall efficiency of converting electricity to hydrogen. Hydrogen transmission energy losses of 0.0022 %/km are assumed based on the Danish Energy Agency's Technology Catalogue. We have further updated our text in the Supplementary material.

Page 8, section 1.05, Supplementary Material:

There are additional operational costs for delivering hydrogen via transmission pipelines. Energy expenses for compression are implicitly accounted for by reducing the overall efficiency of the electrolyzer units by 2.1 %, 1.7 % and 1.5 % for investments made in 2030, 2040, and 2050, respectively. Furthermore, to maintain the operational pressure, hydrogen transmission energy losses of 0.0022 %/km [3] are included. A 4 % discount rate and a lifetime expectancy of 50 years is assumed for the network infrastructure investments. Other costs associated with hydrogen distribution grids and the necessary equipment to supply hydrogen to the consumption sites are not considered.

- SI, p.5 “s. However, only liquid ammonia and methanol are currently competitive for importing without the need for re-conversion “

But most of the hydrogen could be replaced with methanol, e.g. in shipping, for power balancing... At least, there should be a sensitivity analysis towards the impact on the H2 network if all green Ammonia demand is imported.

Please see the previous comment where we addressed these concerns. Furthermore, we have updated our limitations, highlighting the importance of an individual study on multiple importing options.

Page 28, section 1.13, Supplementary Material:

Fourth, we perform a sensitivity analysis to capture the effects of importing hydrogen derivatives such as ammonia, methanol, liquid hydrogen, and other liquid organic hydrogen carriers (LOHC). Future studies should evaluate the competition between various importing fuels via dedicated port terminals, transportation options, and domestic European production. Noted that currently only liquid ammonia and methanol are currently competitive for importing without the need for re-conversion [42].

- SI Figure 3, Figure a and b use same color scale, legend is hiding a number (1e6)

We thank the reviewer for the advice. We have revised accordingly the figures in the Supplementary Material.

- SI p.10 section 1.8 The section should discuss the topic of industry reallocation and possible local shifts in demand.

We thank the reviewer for bringing this to our attention. We believe that potential infrastructure savings and benefits could result from the case industry relocating in proximity to future competitive energy production European regions. After performing the additional modelling with endogenous shifting of hydrogen demand derivatives, we have acknowledged the subject. Already, we have provided insights into the hydrogen network expansion requirements, which may be reduced as a result of ammonia and synthetic fuels demand relocation. Further details have been introduced:

Page 37, section 3.3, Supplementary Material:

We take the opportunity to discuss that the current methodology could be expanded to a general investigation of future European industrial demand relocation, reducing the need for cross-border infrastructure development. Further costs for relocating existing production centers and transportation expenses must be accounted for. Furthermore, social and macroeconomic indicators should also be assessed to provide a holistic approach to the potential benefits.

- SI p.11 “We observe a final demand for hydrogen to be approximately 332 TWh or 10 Mt by 2030 and 1,767 TWh or 53 Mt by 2050. “ Your assumed demand for 2030 is half of the planned demand in REPowerEU (importing 10 Mt H₂ and producing 10 Mt H₂ within Europe). This assumption needs to be justified.

We have added a new section, section 1.11, to the Supplementary Material. There, we examined demand scenarios for hydrogen in Europe. The text indicates that the demand for hydrogen in both the short and long term is uncertain. RePowerEU is one of the most ambitious projections (please see Fig. 8a and 8b). We make use of a demand projection presented by the EHB in our study. These reports illustrate the vision of 28 gas TSOs across Europe. According to our review, the mean projected demand for 2030, as reported by various sources, is 400TWh or 12Mt. This figure closely aligns with the estimates we have made based on the EHB. We have further updated our Main Text as well.

Page 9, section 4.2, Main Text:

Furthermore, we revise the future hydrogen consumption per country in accordance with the European Hydrogen backbone report [5]. A comparison of European hydrogen demand projections can be found in the Supplementary Material, section 1.11.

Page 22, Section 11, Supplementary Material:

Notably, the RePowerEU initiative emerges as a particularly ambitious plan by 2030, with a strong focus on hydrogen integration within the transport sector.

• SI, p.11 “ The report provides country-level hydrogen demand and its penetration into different uses such as ammonia synthesis, liquid fuels and high-value chemicals, high-temperature industrial process heat, and iron ore reduction with direct use of hydrogen. “ This is one of the most critical assumptions in the publication. You assume a fixed hydrogen demand per region which has strong impacts on the resulting hydrogen infrastructure. But e.g. ammonia/high value chemical/ liquid fuels could be produced at other regions and transported then to the demand centers, so the assumption of having a fixed hydrogen demand per region is not correct.

We have previously addressed these concerns by performing additional model runs and assessing several sensitivity scenarios focusing on changes in the demand for hydrogen derivatives and its impact on the optimal infrastructure and network topology.

• SI p.12 Figure 4, label “H₂ demand”
Thank you for your suggestion. We have adapted the figures.

References

Di Lullo, G., Giwa, T., Okunlola, A., Davis, M., Mehedi, T., Oni, A. O., & Kumar, A. (2022). Large-scale long-distance land-based hydrogen transportation systems: A comparative techno-economic and greenhouse gas emission assessment. *International Journal of Hydrogen Energy*, 47(83), 35293-35319.

IRENA. (2022). *Global Hydrogen Trade to Meet the 1.5°C Climate Goal: Technology Review of Hydrogen Carriers*”.

European Commission. (2015). *Assessment of the CO₂ Storage Potential in Europe (CO₂Stop)*.

Danish Energy Agency. (2023). *Technology Data for Carbon Capture, Transport and Storage*.

Eurostat. (2022). *CO₂ emissions from EU territorial energy*.

Siala, K., Mier, M., Schmidt, L., Torralba-Díaz, L., Sheykha, S., & Savvidis, G. (2022). Which model features matter? An experimental approach to evaluate power market modeling choices. *Energy*, 245, 123301.

Caglayan, D. G., Weber, N., Heinrichs, H. U., Linßen, J., Robinius, M., Kukla, P. A., & Stolten, D. (2020). Technical potential of salt caverns for hydrogen storage in Europe. *International Journal of Hydrogen Energy*, 45(11), 6793-6805.

Neumann, F., Zeyen, E., Victoria, M., & Brown, T. (2023). The potential role of a hydrogen network in Europe. *Joule*, 7(8), 1793-1817.

Boitier, B., Nikas, A., Gambhir, A., Koasidis, K., Elia, A., Al-Dabbas, K., ... & Mittal, S. (2023). A multi-model analysis of the EU's path to net zero. *Joule*, 7(12), 2760-2782.

Reviewer #2 (Remarks to the Author):

Title: A unified European hydrogen infrastructure planning to support the rapid scale-up of hydrogen production

Abstract of this paper :

This study examined plausible pathways for hydrogen infrastructure and various methods of hydrogen supply using open-source code model. They identify the effect of blue hydrogen and danger of methane usage for hydrogen production. They also highlight development of electrolysis for self-sufficient hydrogen production.

Although the approach of the paper is interesting, it needs further improvement.

Revision for this paper :

1. The conclusion of the study seems obvious. Please highlight the novelty of the paper.

The study is timely and relevant as European nations have announced medium-term hydrogen strategies, but a pathway towards a unified European hydrogen infrastructure coupled with the remaining energy system is still to be assessed. We thank the reviewer for the careful consideration of our conclusions. We have reformulated the revised section by including additional insights and information. Here, we summarize our contributions.

1) Methodological contribution of performing analysis of pathways instead of examining the energy system only in the final year, as performed in (Neumann et al. <https://doi.org/10.1016/j.joule.2023.06.016>). By conducting pathway assessments, potential lock-in effects can be assessed. Furthermore, we provide insights into how the future hydrogen infrastructure (network and storage) could develop for different plausible scenarios.

Page 8, section 3, Main Text:

In contrast with other studies performing overnight investments by examining only 2050, a brownfield approach is adopted.

2) Methodological and conceptual contributions. We investigate the possibility of hydrogen derivatives demanding spatial relocation and its effects on future European hydrogen network topology and development.

Page 8, section 3, Main Text:

Moreover, we highlight that the potential relocation of demand for hydrogen derivatives located mainly in industrial regions of Germany, the Netherlands, and Belgium towards European countries with competitive hydrogen production can impact the future topology and development of the hydrogen backbone.

3) Conceptual contributions. We investigate the effect on European production centers and infrastructure due to overseas hydrogen derivative imports.

Page 8, section 3, Main Text:

However, in the event where hydrogen derivative demand is met through overseas imports, reaching 500 TWh by 2050, cross-border hydrogen connections could potentially be reduced by 16 % and underground storage requirements reduced by 23 %. It is important to note, though, that the 2030 estimated terminal capacity within the EU is relatively low, approximately 146 TWh/a or 4.4 Mt/a [38].

4) Extended sensitivity analysis on the potential technological changes and natural gas market price signals on the competition of blue, green, and hydrogen imports.

Page 9, section 3, Main Text:

However, this competition is dynamic. We demonstrate that potential technological changes, such as reduced capital expenditures for electrolysis, possibly inadequate carbon capture rates for CCS, and rising costs associated with the storage and transportation of CO₂ or future higher natural gas market prices, could tilt the favor towards green hydrogen.

5) First study to provide insights that when sector coupling benefits and synergies are accounted for, European domestic hydrogen could be competitive to imports from neighbouring countries via dedicated pipelines.

Page 8, section 3, Main Text:

While neighbouring European nations proclaim ambitious hydrogen export goals through dedicated pipelines, we demonstrate that by 2030, domestic hydrogen production is sufficient to meet the anticipated European demand. We show that European hydrogen production can be competitive when sector coupling synergies are considered towards 2050.

2. (Page 2) Please avoid lump sum references. All references should be explained with detailed and specific description.

We have revised our text and avoided lump-sum references. In addition, we have provided more details of relevant studies and updated our literature with the most recent state-of-the-art publications.

Page 2, section 1, Main Text:

Second, while recent studies focus on detailed self-sufficient carbon-neutral European energy system scenarios for 2050, they have not integrated hydrogen networks [9] and focus mainly on the benefits of sector coupling [10]. Third, few recent studies have attempted to use techno-economic optimization methodologies to analyze the value of hydrogen infrastructure in Europe. Some focus on hydrogen grid development in 2050 [11], by assessing multiple weather data [12] or examining both hydrogen and power grid trade-offs [13, 14] yet neglecting pathway dependencies. Fourth, while providing an energy transition pathway analysis, some studies do not include blue hydrogen as a production alternative [15, 16]. Fifth, comprehensive studies are carried out, yet they disregard the interaction of electricity and

hydrogen sectors with the heating sector [17, 18] or while modeling a sector-coupled European energy system, they focus on a specific country or region only. For example, they investigate the role of offshore wind in the North Sea [15], assess if offshore hydrogen production is more competitive than onshore [19], and examine scenarios for designing a net-zero energy system [20] and hydrogen supply pathways to greenhouse gas neutrality in Germany [21]. Finally, some investigate alternative transition pathways for the European energy system, including electricity and hydrogen infrastructure development [22], yet assume self-sufficiency.

3. (page 2) The model used in this study seems old dated. Because hydrogen industry has changed a lot in recent years. Please explain important update list after 2018.

The model presented in this study is a state-of-the-art energy system analysis model. Balmorel has been developed since 2001 by a well-functioning community across international universities. Since 2018, Balmorel has been further developed to encompass full sector coupling of a Pan-European energy system. In line with the development of the hydrogen industry, Balmorel has been extended further to include different hydrogen production pathways and hydrogen infrastructure, including both hydrogen transmission and storage. In the Methods section 4.2, "Balmorel, modeling, and data advancements," we have provided explicit information regarding the most recent model improvement in comparison to the earlier version.

Furthermore, as noted in section 4.1, four other well-known open-source energy system models have been compared to Balmorel with conclusions emphasizing the model's validity (see Candas et al. (2022), <https://doi.org/10.1016/j.rser.2022.112272>, Ouwerkerk et al. (2022), <https://doi.org/10.1016/j.rser.2022.112331>).

The most recent version of the model is available on GitHub (<https://github.com/balmorelcommunity/Balmorel/>), and more information about the model additions be found, e.g., in Bermudez et al. (2023) (<https://doi.org/10.1016/j.enpol.2022.113382>).

In the main text, the Methods section 4.2, we have updated the text and provided detailed information on the modeling and data advancements of the current Balmorel model in comparison with the previous versions.

4. (section 2.2) Although this study analyzed importance of blue hydrogen, supplementary material or explanation of blue hydrogen is insufficient.

We thank the reviewer for bringing up the topic. Firstly, we have added a new section under the Methods to describe how Carbon Capture and Storage (CCS) is modeled in Balmorel. Please see page 10, section 4.4, Main Text.

Furthermore, we have conducted multiple sensitivity analyses to assess the robustness of our main scenario H2E results. Initially, a sensitivity analysis was conducted to examine the impact of reduced

carbon capture rates, given that the majority of operational CCS projects presently demonstrate a capture rate below 90%. Secondly, costs associated with CO₂ storage and transportation remain uncertain. We have conducted additional sensitivity analyses in which the costs were varied. Thirdly, a pessimistic sensitivity analysis was conducted by combining high costs while varying the capture rates. The corresponding section 3.1 is available in the Supplementary Material, page 30. On the basis of these sensitivities, plausible future outcomes are investigated. Even with the most pessimistic sensitivity, blue hydrogen appear; nevertheless, by 2050, it is replaced by green hydrogen. We have updated our statements accordingly in the main text.

Page 7, section 2.7, Main Text:

Carbon capture rate for CCS applications has been estimated to vary from 56 % to 90 % [36]. With low capture rates, SMR technology as an industrial process becomes less competitive due to exposure to the rising EU ETS price, referred to as CO₂ tax in this study. We discover that a low capture rate of 60 % significantly impacts the competition between blue and green hydrogen by 2030, with green hydrogen penetrating at 60 % and accounting for nearly 77 % in 2050 (Extended Data Fig. 6c). Finally, costs for pipeline transportation and CO₂ storage are yet uncertain and can influence the competition between blue and green hydrogen. A recent research study [37], for example, predicts a spread of 3.6 to 41 €₂₀₂₂/tCO₂. According to the sensitivity analysis, a cost of 40 €₂₀₂₂/tCO₂ results in an earlier expansion of green hydrogen with a total output of 160TWh by 2030 and a subsequent increase to 1224TWh by 2050. Finally, a potential combination of high costs (40 €₂₀₂₂/tCO₂) and low capture rates (60 %) can lead to a low blue hydrogen penetration in the short-term years 2030-2040, with blue hydrogen phasing-out of the system by 2050 (see Supplementary Material, section 3).

5. (section 2.4) Hydrogen can be stored in various forms such as LOHC, liquid, and gas. Please consider various hydrogen storage form for hydrogen delivery and storage.

We emphasize that our research focuses on the long-distance, interregional transport of hydrogen between European nations and that the transport form of gas could be the most cost-effective option. Please see our next response, where we analyse the costs of hydrogen transport and storage forms.

6. (section 2.4) Hydrogen can be transported by pipe and tube trailer depending on distance. Please consider the use of various transport method for developing hydrogen pathway.

We thank the reviewer for highlighting this topic and we acknowledge that there are other types of storing and transporting hydrogen. For this reason, we have added a new section in the Supplementary Material, section 1.6, where we review the most recent studies that report costs and alternatives for hydrogen transport and storage forms.

Findings in the literature motivated us not to encompass different forms of hydrogen transportation because they would further complicate the model, while the solution would most likely not change. Recent studies suggest that transporting hydrogen in gas form over long distances via new or repurposed

natural gas pipelines is the cheapest and most efficient method. Moreover, the most recent study, "Global Hydrogen Trade to Meet the 1.5°C Climate Goal: Technology Review of Hydrogen Carriers" IRENA 2022 (see Figure 6.9., page 128), indicates that LOHC could only be a competitive option for small-scale projects, below 0.2 Mth₂/year. Similar insights are presented in Di Lullo et al. (2022), (<https://doi.org/10.1016/j.ijhydene.2022.08.131>), emphasizing that liquid carrier and truck transport is twice as costly as H₂ pipelines at 1000 km, where the cost difference further increases when the distance becomes around 3000km.

The transportation of tube trailers cannot accommodate large volumes at a competitive cost. In contrast, it could be used for small-scale transportation between production and demand sides located at a relatively close distance, which could work well in combination with the cross-regional trade in pipes.

Lastly, we have updated the limitation section in the Supplementary Material.

Page 28, Section 1.13, Supplementary Material:

Fourth, we perform a sensitivity analysis to capture the effects of importing hydrogen derivatives such as ammonia, methanol, liquid hydrogen, and other liquid organic hydrogen carriers (LOHC). Future studies should evaluate the competition between importing various fuels via dedicated port terminals, transportation and storing options, and domestic European production.

7. (section 2.4) In this study, geographical characteristics seems the most important factor for developing hydrogen pathway. The level of technology and policy of countries should be considered for developing the optimal pathway.

We acknowledge that geographical features are significant and may impact the optimal pathway. We already have incorporated electrolysis capacities announced by European nations for 2030 in our scenarios based on national targets (see report: "Clean Hydrogen Europe", Hydrogen Europe 2022). We have provided additional details in the main text.

Page 12, section 7, Main Text:

National electrolysis capacity targets up to 2030 are considered to depict a plausible short-term hydrogen market evolution [4].

As stated in the introduction, we note that the hydrogen strategies declared by European countries are not aligned and largely express national-specific visions for the future European hydrogen economy. In addition, the majority of these strategies are focused only on the year 2030, with no additional objectives for subsequent years.

Regarding the level of technology, as discussed, we have performed a sensitivity analysis on the key parameters of the main hydrogen production technologies (i.e., Electrolysis and SMR). Cost of electrolysis, future projection on natural gas market prices, carbon capture rate, and cost of transporting and storing CO₂.

We note that in section 1.4 of the Supplementary Material, we have discussed the recent delegated act for producing renewable liquid and gaseous transport fuels of non-biological origin (RFNBO). We show that under the high CO₂ tax, our results for 2030 can comply with the delegated acts rules.

Furthermore, to examine if the hydrogen demands assumed in this study are considered in line with other reports and EU targets, we have added a new section 1.11, where we review European hydrogen demand projections (see Fig. 10). There, we note that the most recent targets announced in the RePowerEU are considered to be the most ambitious. Our demands (i.e., Guidehouse EHB) for 2030 are in line with the average projections.

Page 22, section 1.11, Supplementary Material:

The projections show minor discrepancies for the year 2030 (Fig. 10a) but diverge significantly for 2050 (Fig. 10b). Notably, the RePowerEU initiative emerges as a particularly ambitious plan by 2030, with a strong focus on hydrogen integration within the transport sector.

8. (section 2.7) In case of Japan and south Korea, actually production price of hydrogen gas is continuously increasing. The government is paying subsidies to fall the hydrogen price for market. Please explain the reason for hydrogen price reduction.

We have modified the text in section 2.7 based on the reference [31] providing more information on the potential decrease costs of hydrogen production. For example, Ref. 31 "Green hydrogen cost reduction" by IRENA 2020 emphasizes that technological advancements, such as stack design, enable increased catalyst surface area and utilization increasing future electricity to hydrogen efficiency. In addition, economies of scale can reduce the cost of electrolysis with larger module sizes for commercialization and larger production facilities. Material, labour, capital, standardization and energy costs are anticipated to decrease as well. Lastly, the report emphasizes that "learning by doing" can decrease future expenses.

We appreciate the reviewer's example, and due to other concerns introduced in ref. [33], such as short-term scarcity of material and infrastructural availability and long-term uncertainty, the rate of the aforementioned technical and economic improvements, and system costs may not be realized. These factors prompted us to conduct a sensitivity analysis (see section 2.7) in which we increased the anticipated electrolysis CAPEX and observed differences in production pathway competition.

Page 12, section 7, Main Text:

Recent research [30] based on market surveys for alkaline electrolyzer technology reveals a significant spread between projected capital expenditures in 2030 and 2040. However, according to international organizations, there is widespread agreement that the cost of producing renewable hydrogen will continue to fall toward 2050 [31, 32] due to technological advancements and learning by doing [33]. Yet there are still lingering concerns about this assumption, such as short-term scarcity of material and infrastructural availability [34].

9. (supplementary material) Although many countries proclaimed ambitious goals, achievement is unsatisfactory. To develop realistic hydrogen pathway, the model should consider real achievement rate of country's roadmap."

We have considered the national strategies up to 2030 for electrolysis capacities. For future years, we let the model optimize the energy mix. We provide the starting point as realistic as possible. In addition, in section 2 of the Supplementary material, we have compared our electrolysis investment rates with those of other studies employing various methodologies, such as probabilistic econometrics or other 2050-focused energy system analysis studies. These confirms that our rates could be feasible. Furthermore, we have added another paragraph comparing our renewable energy investment results with other studies and EU targets.

Page 28, section 2.1, Supplementary Material:

The recent 2022 EU market outlook for solar power [43] reveals a significant surge of nearly 50% percent year-over-year in 2022, a record of 41.4 GW solar PV additional generation capacity. This development indicates that our near-term projections for 2030 (H2E scenario 55 GW) could be achieved, assuming similar increasing growth rates. However, our projections diverge from the European Commission's offshore wind power generation vision, which seeks to establish 300 GW of capacity by 2050 [44]. In the H2E scenario, we find 153 GW, whereas, in the GH2E scenario, a future system without blue hydrogen requires 213 GW offshore wind. When compared to other European system studies, a similar high renewable penetration is discovered. For example, Neumann et al. [45], by 2050, project solar PV installations ranging from 2666 to 3598 GW, onshore wind from 1691 to 1776 GW, and offshore wind from 206 to 245 GW. Another study [23] assessing 2050 carbon neutrality scenarios estimates solar PV ranges from 2146 to 2449 GW, onshore from 639 to 726 GW, and offshore from 201 to 227 GW.

10. (section 3) Rapid scale-up of electrolysis capacity is important, but it needs enormous investment. The model should consider current development situation of each technology.

We acknowledge that the investments in the green hydrogen value chain are enormous. Based on the most recent report published by the International Energy Agency (IEA) in 2023 titled "Global Hydrogen Review 2023", we updated the content in section 2.3 of the Supplementary material on the present development of electrolysis. The report reveals that the present European electrolysis fleet is approximately 0.5GW, whereas REpowerEU's goals for 2030 call for at least 60GW. Furthermore, recent work by Odenweller et al. applying probabilistic techniques for projecting electrolysis capacity deployment in Europe verifies our energy system analysis projections. Our scenarios could be plausible if the electrolysis deployment rates increased.

Page 30, section 2.3, Supplementary Material:

By comparing our electrolysis capacity growth to a recent study by Odenweller et al. [50], we illustrate that our projections for 2030 and 2050 are feasible in the scenarios. In addition, we emphasize that the

SSGH2E scenario is consistent with the European Union's Hydrogen strategy (500 GW [50]) by 2050. We witness an electrolysis fleet with an installed capacity of 507 GW, which supplies a total hydrogen production of 1767 TWh. Another recent study [51] illustrates plausible pathways, towards 2050, for electrolysis deployment in the European system ranging from 1378 GW to 2186 GW, satisfying a total hydrogen demand of approximately 3150TWh. However, this demand is nearly twice as high as indicated by this study. According to the latest IEA Global Hydrogen Review 2023 [52], the current capacity of the European electrolysis fleet stands at approximately 0.5 GW. However, ambitious targets exceeding 60 GW [53] have been announced in the RePowerEU [54] initiative. Our 2030 results range from 24-68 GW depending on the scenario.

References

Candas, S., Muschner, C., Buchholz, S., Bramstoft, R., van Ouwkerk, J., Hainsch, K., ... & Justin, A. (2022). Code exposed: Review of five open-source frameworks for modeling renewable energy systems. *Renewable and Sustainable Energy Reviews*, 161, 112272.

van Ouwkerk, J., Hainsch, K., Candas, S., Muschner, C., Buchholz, S., Günther, S., ... & Bramstoft, R. (2022). Comparing open source power system models-A case study focusing on fundamental modeling parameters for the German energy transition. *Renewable and Sustainable Energy Reviews*, 161, 112331.

Gea-Bermúdez, J., Bramstoft, R., Koivisto, M., Kitzing, L., & Ramos, A. (2023). Going offshore or not: Where to generate hydrogen in future integrated energy systems?. *Energy Policy*, 174, 113382.

IRENA. (2022). *Global Hydrogen Trade to Meet the 1.5°C Climate Goal: Technology Review of Hydrogen Carriers*".

Di Lullo, G., Giwa, T., Okunlola, A., Davis, M., Mehedi, T., Oni, A. O., & Kumar, A. (2022). Large-scale long-distance land-based hydrogen transportation systems: A comparative techno-economic and greenhouse gas emission assessment. *International Journal of Hydrogen Energy*, 47(83), 35293-35319.

Hydrogen Europe. (2022). *Clean Hydrogen Europe*.

International Energy Agency (IEA). (2023). *Global Hydrogen Review 2023*.

REVIEWER COMMENTS

Reviewer #1 (Remarks to the Author):

The paper has been greatly improved in the revised version. Modelling assumptions are described more clearly in the new version. In addition, the authors have made an impressive effort to check the robustness of the results through further sensitivity analyses.

There are still some points that should be clarified:

Demand shift

- "We run multiple sensitivity analyses so that up to 60% of the synthetic fuel demand can shift."

-> Why did you choose 60% and not 100%?

-> Please describe in the text, how much does your network topology change (e.g. change in pipe capacity, transported hydrogen) if the demand can be shifted spatially?

H2 Imports

- "Nonetheless, a sensitivity analysis regarding prospective imports of hydrogen derivatives from overseas reveals a reduction in the deployment of hydrogen network and underground storage. For example, in the H2E, a potential scenario involving overseas derivative imports of 500TWh by 2050 could lead to a 32% reduction in the hydrogen network's relative expansion (see Supplementary Material, section 3)."

-> Please set into context: How does this relate to planned political targets? E.g. how large are the imports of hydrogen derivatives compared to political targets?

- "Regarding the optimal transportation of hydrogen or derivatives, in SM section 1.6, we performed a literature review. We elaborated further on the cost differences and competitiveness between pipelines and other means for transporting hydrogen or derivatives. Recent studies show that hydrogen pipelines are the most competitive option over long distances compared to others, such as ammonia pipelines or ships, LOCH pipelines or rails, and hydrogen trucks in gas or liquid form. (Di Lullo et al. 2022

<https://doi.org/10.1016/j.ijhydene.2022.08.131>, report: "Global Hydrogen Trade to Meet the 1.5°C Climate Goal: Technology Review of Hydrogen Carriers" IRENA 2022). "

-> There are already a few LNG places which should be retrofitted to import green hydrogen or derivatives. Especially for some chemical carriers, e.g. methanol, transport via pipeline is not necessarily the cheapest option [1], this should be added to the discussion since imports via LNG terminals are planned already.

[1] <https://www.ise.fraunhofer.de/en/press-media/press-releases/2023/fraunhofer-ise-study-where-germanys-imports-for-hydrogen-and-power-to-x-products-could-come-from.html>

CO₂ management

-> How large is the CO₂ storage potential? There are different estimates in the CO₂StoP database

-> How much CO₂ is stored in the different investment periods? How does this compare to other studies/ the Net Zero Industry Act you are mentioning?

-> Please indicate somewhere your CO₂ price estimates for each investment period. What is the source for your assumptions?

Hydrogen modelling

- "Our results shed light on the competition between gray, blue, and green hydrogen production pathways. Gray hydrogen is outcompeted based on a high CO₂ taxation projection (150 e/tCO₂, by 2030) implemented across the scenarios."

-> so the existing SMR capacities are stranded assets since not all of them will be at the end of their lifetime? This should be mentioned in the text.

- "Page 8, section 3, Main Text:

However, in the event where hydrogen derivative demand is met through overseas imports, reaching 500 TWh by 2050, cross-border hydrogen connections could potentially be reduced by 16 % and underground storage requirements by 23 %. It is important to note, though, that the 2030 estimated terminal capacity within the EU is relatively low, approximately estimated as 146 TWh/a or 4.4 Mt/a [38].

Moreover, we highlight that the potential shifting of hydrogen derivatives demand located mainly in industrial regions of Germany, the Netherlands, and Belgium towards European countries with competitive hydrogen production can impact the future topology and development of the hydrogen backbone."

-> As mentioned in my previous comment in the first review, did you compare to a case without any hydrogen network? How would that impact the total system costs?

-> You consider onshore hydrogen storage. There are many environmental concerns about onshore hydrogen storage connected to highly saline water which potentially is harmful for the environment and could lead to potential soil or groundwater contamination. This should be discussed as a limitation in the Supplementary Material.

Binding constraints

-> is there a build-out constraint of electrolysis or renewables? Are these constraints binding? If there are binding constraints, it should be mentioned when describing the results.

-> how much did the electricity network expand after 2035?

Code

-> which branch was used for this study? Is there an up to date documentation? what was added for this study to the model in the code base?

Minor points

-> spelling should be checked in some of the added parts

e.g. p.5 missing spaces "500TWh" or p.10 wrong punctuation "generating electricity. hydrogen and heat. "

Reviewer #1 (Remarks on code availability):

It was not clear to me which branch was used. I asked this question to the authors.

Reviewer #2 (Remarks to the Author):

All responses are well prepared.

Reviewer #2 (Remarks on code availability):

Since the model used open-sources energy system model, the results is reproducible, and have validity.

Detailed Responses to Reviewers:

We would like to thank the two reviewers for their constructive and useful feedback. These were very helpful, and we were able to address all of the comments and adapt the submission accordingly. We have copied the entire set of comments below, with our reply in blue and the included paragraphs in italics, which are also noted in the track-changes version of the main manuscript in red colour.

Reviewer #1 (Remarks to the Author):

The paper has been greatly improved in the revised version. Modelling assumptions are described more clearly in the new version. In addition, the authors have made an impressive effort to check the robustness of the results through further sensitivity analyses.

The authors would like to thank the reviewer for the well-structured feedback and recommendations on improving the quality of our manuscript. We greatly appreciate the time and effort you invested in reviewing our work.

There are still some points that should be clarified:

Demand shift

- "We run multiple sensitivity analyses so that up to 60% of the synthetic fuel demand can shift."
- > Why did you choose 60% and not 100%?

The sensitivity analysis on the hyperparameter gamma was conducted to demonstrate the robustness of the hydrogen grid development due to the spatial uncertainty of the demand for synthetic fuels. A potential 100% assumption will correspond to a massive demand shift of renewable fuel from one to another model region, which, in reality, might be less likely. However, given the nature of the analyses as explanatory, we have performed additional runs up to 100%. We have updated the figures (as mentioned in the following) and provided additional insights based on the supplementing results.

In the case of an H2E scenario in which blue hydrogen may lock in, the influence of a probable 100% shift in synthetic fuels has little impact on total hydrogen network expansion compared to the prior 60% results (see Supplementary Fig. 27a). The optimal results illustrate that major demand centers with competitive renewable resources have a high potential to receive demand, especially when they are located close to regions with a tendency to decrease synthetic fuel demand, such as the Netherlands, Belgium and German regions. One example is France (see Supplementary Fig. 26). Furthermore, the rest of the derivatives and direct hydrogen demands can be met by local production from SMR-CCS units.

In contrast, in a future system where green hydrogen predominates, we further verify our previous hypothesis that a change in demand might significantly impact the growth of hydrogen infrastructure (see Fig. 27b). For instance, a 28% hydrogen network decline resulted from a 100% change in demand relocation in 2050, and a 20% produced from a 60% shift.

Page 47, section 4.4, Supplementary Information:

Across the two scenarios, we observe similar resulting trends. The large production centers located at the periphery of Europe attract significant demand for hydrogen derivatives (Supplementary Fig. 26). Furthermore, industrial regions of Germany and nations such as Belgium and the Netherlands fully uti-

lize the available shifting potential, relaxing the network development expansion requirements as captured in Fig. 27a and 27b. The corridors from the United Kingdom and Ireland or from Spain and France appear significantly smaller. In addition, a sensitivity analysis of the flexibility factor parameter θ reveals that demand allocation can significantly impact the European network, particularly if blue hydrogen is unavailable (Fig. 19a and 19b). For example, in the case of the GH2E scenario, a 60 % available shift could result in a 20 % lower Pan-European network development by 2050, while a 100 % shift results in a 28 % decrease.

Finally, in section 4.4 at the SI, we already acknowledged the limitations (e.g., transportation costs of final fuel to end consumers) of our analysis. The benefits of co-producing hydrogen and its derivatives reduce cross-border hydrogen pipelines and challenge the design of the envisioned European Hydrogen Backbone when the spatial demand for hydrogen derivatives is uncertain. We have also highlighted that the derivative's additional transportation costs to the final consumption site are not considered. We also discuss potential industrial demand relocation, which should be assessed through a comprehensive modeling approach outside the research scope of this study.

Page 47, section 4.4, Supplementary Information:

We take the opportunity to discuss that the current methodology could be expanded to a general investigation of future European industrial demand relocation, reducing the need for cross-border infrastructure development. Further costs for relocating existing production centers and transportation expenses must be accounted for. Furthermore, social and macroeconomic indicators as well as national industry policies should also be assessed to provide additional insights to the potential benefits.

-> Please describe in the text, how much does your network topology change (e.g. change in pipe capacity, transported hydrogen) if the demand can be shifted spatially?

We have updated the Main Text and provided the related information.

Page 8, Hydrogen corridors and storage, cross-regional trading, Main Text:

Another mechanism for reducing or altering the topology of future hydrogen networks could be the co-location of hydrogen production and demand for derivative fuels. We demonstrate that network expansion could be reduced by 16 % in 2050 if the demand for 613 TWh hydrogen derivatives is not spatially fixed in the GH2E scenario. Furthermore, the new network development reduces the volume of imported hydrogen to countries like Germany, the Netherlands, and Belgium, decreasing from 300 TWh in the GH2E scenario to 141 TWh with 60% spatial demand flexibility. Our analysis challenges the design of the envisioned European Hydrogen Backbone, which might be revised in light of spatial uncertainty in the demand for hydrogen derivatives (see Supplementary Information, section 4.4).

H2 Imports

- "Nonetheless, a sensitivity analysis regarding prospective imports of hydrogen derivatives from overseas reveals a reduction in the deployment of hydrogen network and underground storage. For example,

in the H2E, a potential scenario involving overseas derivative imports of 500TWh by 2050 could lead to a 32\% reduction in the hydrogen network's relative expansion (see Supplementary Information, section 3)."

-> Please set into context: How does this relate to planned political targets? E.g. how large are the imports of hydrogen derivatives compared to political targets?

We appreciate the reviewer's observations. The European Commission has yet to formally declare any political objectives for imports of hydrogen and derivatives at the EU level by the year 2050. We reflected on our results, taking into account the most current RePowerEU import objective (10 million tons/a) and the expected announced projects for overseas hydrogen derivative imports to reflect the potential savings from hydrogen infrastructure development (network and storage). We modified our main text as follows:

Page 10, Discussion and conclusion, Main Text:

However, in the event where hydrogen derivative demand is met through overseas imports, reaching 500 TWh by 2050, cross-border hydrogen connections could potentially be reduced by 16% and underground storage requirements by 23%. Notably, the 2030 estimated terminal capacity for hydrogen derivatives within the EU is approximately 146 TWh/a or 4.4 Mt/a [39]. In contrast, RePowerEU anticipates hydrogen and derivatives imports of 10 Mt/a by 2030, indicating that expanding port infrastructure is necessary to offset potential savings from a reduced hydrogen network and storage development.

- "Regarding the optimal transportation of hydrogen or derivatives, in SM section 1.6, we performed a literature review. We elaborated further on the cost differences and competitiveness between pipelines and other means for transporting hydrogen or derivatives. Recent studies show that hydrogen pipelines are the most competitive option over long distances compared to others, such as ammonia pipelines or ships, LOCH pipelines or rails, and hydrogen trucks in gas or liquid form. (Di Lullo et al. 2022 <https://doi.org/10.1016/j.ijhydene.2022.08.131>, report: "Global Hydrogen Trade to Meet the 1.5°C Climate Goal: Technology Review of Hydrogen Carriers" IRENA 2022). "

-> There are already a few LNG places which should be retrofitted to import green hydrogen or derivatives. Especially for some chemical carriers, e.g. methanol, transport via pipeline is not necessarily the cheapest option [1], this should be added to the discussion since imports via LNG terminals are planned already.

[1] <https://www.ise.fraunhofer.de/en/press-media/press-releases/2023/fraunhofer-ise-study-where-germanys-imports-for-hydrogen-and-power-to-x-products-could-come-from.html>

We thank the reviewer for supplying the most recent Fraunhofer-ISE study. We have already acknowledged in the main text and Supplementary Information (please look at the previous answer) regarding the future plans for overseas import via port terminals. In addition, we have added to the discussion and further elaborated on our text regarding hydrogen transportation means, stressing the findings from the Fraunhofer-ISE study. The report, makes the case for competitive derivative imports from nations like

Brazil, Colombia, and Australia and supporting competitive gaseous hydrogen imports from North Africa if the infrastructure is developed.

Page 11, section 1.6, Supplementary Information:

This suggests that leveraging the potential lower hydrogen production costs in North Africa [13, 18], coupled with the cost advantages of repurposing existing natural gas pipelines, position North African gas hydrogen imports as a competitive import option for Europe. A recent study [19] assessing site-specific gas hydrogen and derivatives import costs further supports the above conclusion, provided that the hydrogen network will be available in time for transport. The study further emphasizes that countries like Brazil, Colombia, and Australia might offer competitive import conditions for liquid hydrogen, ammonia, and methanol. Repurposing or building new fuel infrastructure in ports would, however, need to be taken into account to assess the competitiveness of that solution, which was outside the scope of this study.

CO₂ management

-> How large is the CO₂ storage potential? There are different estimates in the CO₂StoP database

-> How much CO₂ is stored in the different investment periods? How does this compare to other studies/ the Net Zero Industry Act you are mentioning?

We appreciate the reviewer's observation. We have acknowledged the targets of the NetZero Industry Act for 2030 and 2050, please see below.

Page 13, Carbon capture and storage, Main Text:

The European Commission recently published the Net Zero Industry Act [66], highlighting that a key bottleneck for carbon capture investments is the lack of operating CO₂ storage sites. The European Commission sets a Union target of 50 Mt of annual operational CO₂ injection capacity by 2030 with a potential estimate of 550 Mt by 2050 [66].

The total theoretical capacity for all countries assessed by CO₂StoP (see Anthonen and Christensen 2021) was estimated to be 625 Gt CO₂. An updated study from the Clean Air Task Force 2023 (Report: Unlocking Europe's CO₂ Storage Potential Analysis of Optimal CO₂ Storage in Europe) provides a theoretical potential between 262 and 1520 Gt CO₂. Consequently, the theoretical potential is considered large. Furthermore, in our limitation section, we discussed the potential benefits of a CO₂ network and storage infrastructure, which, to our knowledge, is an open topic in the European sector coupled energy system models providing optimal pathways.

Page 30, section 1.13, Supplementary Information:

Third, the current study implicitly models CO₂ transport and storage. Future research should explore the integration of CO₂ network grids and storage within European contexts [43], yet through a sector-coupled approach. This approach could reveal additional benefits, notably in the production of renewable biogenic liquid fuels, especially considering regions with sustainable biomass potential.

However, we believe that in the early years, CO2 storage resources may be scarce, which is one of the reasons motivating us for the second scenario GH2E (i.e, not allowing blue hydrogen). In addition, we have already performed sensitivity analysis on the 1) cost of transporting and storing and 2) the effect of capture rates deemed less than 90% and a combination of those two (see main text, section: “Future technological advancement and fuel price can impact the hydrogen production pathway”).

To further assess the robustness of our early-year results, we conduct a sensitivity analysis in which we limit the potential for CO2 storage over the examined optimization years based on the announced capacity of projects up to 2030 (Tumara et al. 2024, see Annex 1, Onshore + Offshore storage announced projects). We perform a linear interpolation on the announced storage projects for the years 2035-2050. Table 1 illustrates the amount of CO2 captured and stored in our H2E scenario, primarily captured from hydrogen and electricity production, and compares it to values from the linear interpolations of current and future projects. The implementing constraint is non-binding. However, we note that with Balmorel, we have yet to capture all of the potential CCS uses (industrial and non-industrial activities) that will compete for CO2 storage capacity in the near future. Because the competition is unclear, we do several sensitivity analyses on the potential shown in Table 1, ranging from 60% to 20%. The findings show that green hydrogen is promoted when the constraint is binding, which primarily influences the early years of deployment rather than the long term.

Table 1: Carbon Captured and stored at System level, H2E scenario. Announced projects for 2030 and future projections of CO2 storage. Units: Mt

	2030	2035	2040	2045	2050
H2E	124.96	198.51	242.06	254.80	254.83
Projects	172.83	314.22	476.09	637.96	799.83

We have updated the main text as follows and added the new analysis in the supplementary information section 4.2, pages 41-42.

Page 10, Future technological advancements and fuel price can impact the hydrogen production pathway, Main Text:

In the coming years, industrial and non-industrial CCS applications will compete for limited EU CO2 storage projects [38]. If the expansion of operational storage facilities does not meet the demand between 2030 and 2040, the deployment of blue hydrogen could be decreased. Such a scenario would prioritize the development of green hydrogen and promote imports, demanding an earlier expansion of the hydrogen infrastructure as determined in the H2E scenario. For further analysis of limited CO2 storage project development, see Section 4.2 in the Supplementary Information.

-> Please indicate somewhere your CO2 price estimates for each investment period. What is the source for your assumptions?

We have included the data assumptions in the Supplementary Information.

Page 51, section 5.1, Supplementary Information:

Following the Net Zero Emissions by 2050 (NZE) scenario, the CO₂ quota price projections (see Table 11) are extracted from the World Energy Outlook (WEO 2022) [56]. The trend relates to countries within the Organisation for Economic Cooperation and Development (OECD).

Hydrogen modelling

- "Our results shed light on the competition between gray, blue, and green hydrogen production pathways. Gray hydrogen is outcompeted based on a high CO₂ taxation projection (150 e/tCO₂, by 2030) implemented across the scenarios."

-> so the existing SMR capacities are stranded assets since not all of them will be at the end of their lifetime? This should be mentioned in the text.

We thank the reviewer for the accurate remark. We have updated the text as follows:

Page 5, Hydrogen production centers: Short-term to long-term perspective, Main text:

Our results shed light on the competition between gray, blue, and green hydrogen production pathways. Gray hydrogen is out-competed based on the CO₂ taxation projection (150 €/tCO₂, by 2030) implemented across the scenarios. Consequently, existing conventional SMR capacities may become stranded assets before their operational lifetimes are reached.

- "Page 8, section 3, Main Text:

However, in the event where hydrogen derivative demand is met through overseas imports, reaching 500 TWh by 2050, cross-border hydrogen connections could potentially be reduced by 16 % and underground storage requirements by 23 %. It is important to note, though, that the 2030 estimated terminal capacity within the EU is relatively low, approximately estimated as 146 TWh/a or 4.4 Mt/a [38].

Moreover, we highlight that the potential shifting of hydrogen derivatives demand located mainly in industrial regions of Germany, the Netherlands, and Belgium towards European countries with competitive hydrogen production can impact the future topology and development of the hydrogen backbone."

-> As mentioned in my previous comment in the first review, did you compare to a case without any hydrogen network? How would that impact the total system costs?

We appreciate the reviewer's remark. We conducted the analysis by limiting the hydrogen network investment to the two main scenarios, H2E and GH2E. Similar to Neumann et al., 2023 (i.e., +1.6%), the GH2E scenario's overall system costs increased by 1.49% for 2050, but the H2E scenario's cost difference is 0.95%. Based on this comparison, we have updated the Supplementary Information, Comparison with other studies section.

Page 37, section 3.3, Supplementary Information:

A scenario where hydrogen networks are not allowed, like Neumann et al., results in a similar magnitude of total system cost differences by 2050 of +1.49% and 0.95% for GH2E and H2E, respectively.

-> You consider onshore hydrogen storage. There are many environmental concerns about onshore hydrogen storage connected to highly saline water which potentially is harmful for the environment and could lead to potential soil or groundwater contamination. This should be discussed as a limitation in the Supplementary Information.

We have elaborated further on the potential limitations of using onshore underground storage for hydrogen as follows:

Page 30, section 1.13, Supplementary Information:

Notably, a potential degraded integrity of caprock in onshore underground hydrogen storage may lead to the contamination of underground water sources [41], limiting the possibility of onshore hydrogen storage.

Binding constraints

-> is there a build-out constraint of electrolysis or renewables? Are these constraints binding? If there are binding constraints, it should be mentioned when describing the results.

We have not explicitly constrained the building out of electrolysis or renewables. However, we note that the hydrogen demand, for example, implicitly limits the build-out of electrolysis capacities. Furthermore, we have included a section in the Supplementary Information section that compares the study outcomes, which include mostly renewable growth rates, blue hydrogen development, and electrolysis investments, to those of other studies. Please look at:

Page 36-37, Supplementary Information, Sections 3.1 – 3.3

-> how much did the electricity network expand after 2035?

We elaborated on the power grid reinforcement. Please see Supplementary Figure 5.

Page 12-13, section 1.8, Supplementary Information:

Lastly, Supplementary Fig. 5 illustrates the optimized network development. The network expansion is limited to TYNDP projections for 2035 until that year (section: Scenario choice and description, Main Text). The model then determines optimal capacities but is limited to a total additional capacity of up to 10 GW per cross-border line connection. Compared to 2020, the results indicate that the electricity network could grow by a factor of four by 2050. Minor variations are noted among the scenarios.

Furthermore, in the main text section- “Scenario choice and description”, we provided an extended discussion of our scenario design assumptions for power cross-border reinforcement relating to TYNDP project delays and other constraints restricting potential future interconnections.

Code

-> which branch was used for this study? Is there an up to date documentation? what was added for this study to the model in the code base?

We are upgrading the official Balmorel documentation to reflect the most recent status. This study thoroughly describes new modeling developments (see Main Text, Methods) and expands on modeling assumptions in the Supplementary Information. The study framework is publicly available in the following GitHub branch (Git Branch), and a Zenodo repository (Zenodo Link) has been established to accompany the paper.

We have updated the section on code and data availability.

Minor points

-> spelling should be checked in some of the added parts
e.g. p.5 missing spaces "500TWh" or p.10 wrong punctuation "generating electricity. hydrogen and heat."
"

We appreciate the reviewer's attentive consideration of our work and contributions to improving the quality of our manuscript. We examined the material thoroughly and corrected all formatting errors.

Reviewer #1 (Remarks on code availability):

It was not clear to me which branch was used. I asked this question to the authors.

Please look at our previous response.

References

Anthonsen, K. L., & Christensen, N. P. (2021). *EU Geological CO2 storage summary. Prepared by the Geological Survey of Denmark and Greenland for Clean Air Task Force [Revised, October 2021]*. GEUS. Danmarks og Grønlands Geologiske Undersøgelse Rapport Bind 2021 Nr. 34 <https://doi.org/10.22008/gpub/34594>

Clean Air task force. (2023). *Unlocking Europe's CO2 Storage Potential Analysis of Optimal CO2 Storage in Europe. Report.* <https://www.catf.us/resource/unlocking-europes-co2-storage-potential-analysis-optimal-co2-storage-europe/>

Tumara, D., Uihlein, A. and Hidalgo Gonzalez, I., Shaping the future CO2 transport network for Europe, Publications Office of the European Union, Luxembourg, 2024, doi:10.2760/582433, JRC136709.

Neumann, F., Zeyen, E., Victoria, M., & Brown, T.. The potential role of a hydrogen network in Europe. *Joule*, 7(8), 1793-1817. <https://doi.org/10.1016/j.joule.2023.06.016>

Reviewer #2 (Remarks to the Author):

All responses are well prepared.

Reviewer #2 (Remarks on code availability):

Since the model used open-sources energy system model, the results is reproducible, and have validity.

We thank the reviewer for the positive feedback on our manuscript, particularly regarding the preparation of our responses. We greatly appreciate the time and effort you invested in reviewing our work.

REVIEWERS' COMMENTS

Reviewer #1 (Remarks to the Author):

All my points are answered.

Reviewer #1 (Remarks on code availability):

The link to the Balmorel homepage at the github page (<https://github.com/balmorelcommunity/Balmorel/tree/A-unified-European-hydrogen-infrastructure-planning-to-support-the-rapid-scale-up-of-hydrogen-production>) seems to be broken (links to <https://balmorel.com/>). The repository has a readme, it would be nice to link there to the documentation pdf on how to get started (<https://github.com/balmorelcommunity/Balmorel/blob/A-unified-European-hydrogen-infrastructure-planning-to-support-the-rapid-scale-up-of-hydrogen-production/base/documentation/BalmorelGettingStarted-BGS302.pdf>). The code for the paper is available on github. I couldn't run the model since I don't have a GAMS license.

Detailed Responses to Reviewers:

We would like to thank the two reviewers for their constructive and useful feedback during the review process. These were very helpful and improved the quality of our submission accordingly.

Reviewer #1 (Remarks to the Author):

Reviewer #1 (Remarks to the Author):

All my points are answered.

We thank the reviewer for the positive feedback on our manuscript, particularly regarding the preparation of our responses. We greatly appreciate the time and effort you invested in reviewing our work.

Reviewer #1 (Remarks on code availability):

The link to the Balmorel homepage at the github page (<https://github.com/balmorelcommunity/Balmorel/tree/A-unified-European-hydrogen-infrastructure-planning-to-support-the-rapid-scale-up-of-hydrogen-production>) seems to be broken (links to <https://balmorel.com/>). The repository has a readme, it would be nice to link there to the documentation pdf on how to get started (<https://github.com/balmorelcommunity/Balmorel/blob/A-unified-European-hydrogen-infrastructure-planning-to-support-the-rapid-scale-up-of-hydrogen-production/base/documentation/BalmorelGettingStarted-BGS302.pdf>). The code for the paper is available on github. I couldn't run the model since I don't have a GAMS license.

We have tested the link and found that it correctly redirects to the appropriate Github branch applied to this study. Additionally, the model with the input data can be accessed via the Zenodo repository accompanying this study. Thank you for the recommendation. We have now updated the repository's README (i.e., LINK) to point directly to the documentation PDF.